# New explicit formulae for the settling speed of prolate spheroids in the atmosphere: theoretical background and implementation in AerSett v2.0.2.

Sylvain Mailler[1,3], Sotirios Mallios[2], Arineh Cholakian[1], Vassilis Amiridis[2], Laurent Menut[1], and Romain Pennel[1]

[1]LMD/IPSL, École Polytechnique, Institut Polytechnique de Paris, ENS, PSL Research University, Sorbonne Université, CNRS, Palaiseau France
[2]National Observatory of Athens (IAASARS), Athens, Greece
[3]École des Ponts-ParisTech, Marne-la-Vallée, France

**Correspondence:** Sylvain Mailler (sylvain.mailler@lmd.ipsl.fr)

**Abstract.** We propose two explicit expressions to calculate the settling speed of solid atmospheric particles with prolate spheroidal shapes under the hypothesis of horizontal and vertical orientation. The first formulation is based on theoretical arguments only. The second method, valid for particles with mass-median diameter up to $1000\mu m$, is based on recent heuristic drag expressions based on numeric simulations. We show that these two formulations show equivalent results within 2% for $d_{eq} \leq 100\,\mu m$, and within 10% for particles with $d_{eq} \leq 500\,\mu m$ falling with a horizontal orientation, showing that the first, more simple method is suitable for virtually all atmospheric aerosols, provided their shape can be adequately described as a prolate spheroid. Finally, in order to facilitate the use of our results in chemistry-transport models, we provide an implementation of the first of these methods in AerSett v2.0.2, a module written in Fortran.

## 1 Introduction

Mineral dust plays an important role in the Earth's atmosphere, and in the Earth System in total, influencing radiation, precipitation, and biochemical processes. The impact of dust on each of these processes depends strongly on its Particle Size Distribution (PSD). In terms of radiation, fine dust particles (with sizes less than $5\,\mu m$) scatter the solar radiation, leading to a cooling effect on the global climate, while coarse particles (sizes larger than $5\,\mu m$) tend to absorb both solar and thermal radiation, leading to global warming (Kok et al., 2017). Regarding the precipitation process, dust particles interact with liquid or ice clouds by acting as nucleating particles (Creamean et al., 2013; DeMott et al., 2003; Marinou et al., 2019; Solomos et al., 2011; Twohy et al., 2009). In principle, larger particles are more efficient condensation nuclei, but the number of particles is also an important parameter, and thus the number of particles above a critical size is the quantity that regulates the process (Dusek et al., 2006). Finally, the amount of deposited mass on ocean and land, regulated by the large particles, can stimulate the biochemical activity (Jickells et al., 2005).

The PSD vary greatly over space and time after its emission, since the size dependent process of the gravitational settling removes large particles faster than small particles (e.g. Seinfeld and Pandis, 2006). There is still a large discrepancy between

observations and results produced by transport models regarding the evolution of dust particle lifecycle. Although several observation studies have shown that particles with sizes larger than $30\,\mu m$ can be transported in the atmosphere for days, covering a distance of several thousand kilometers (Goudie and Middleton, 2001; Denjean et al., 2016; Weinzierl et al., 2017; van der Does et al., 2018), several comparisons between model simulations and measurements show that models overestimate the large particles removal (e.g. Ginoux et al., 2001; Colarco et al., 2002). As a matter of fact, the mass of coarse particles in the atmosphere is estimated to be four times larger than the simulated by climate models (Adebiyi and Kok, 2020). All these signify the importance of proper modeling of the mineral dust transport.

The most important force that appears in the dynamics of dust particles, modifying significantly their settling velocity is the drag force. The majority of dust transport models use the Stokes (1851) formulation for the quantification of the drag force (or equivalently the drag coefficient), since they represent mainly spherical particles that are smaller than $20\,\mu m$ (Kok et al., 2021). For larger particles, a correction must be applied to take into account the deviation from the creeping flow, but all the corrections are based on empirical data and can lead to significant differences in the calculated settling velocity (Goossens, 2019; Adebiyi et al., 2023). According to a benchmark between different drag coefficient parameterizations suitable for spherical particles of all natural aerosol and particle sizes presented by Goossens (2019), it has been found that the empirical drag coefficient derived by Clift and Gauvin (1971) seems to perform better than all the others.

Drakaki et al. (2022) used the drag coefficient expression of Clift and Gauvin (1971) in the GOCART–AFWA dust scheme of WRFV4.2.1, and managed to increase the simulated size of dust particles from $20\,\mu m$ to $100\,\mu m$. In the case of coarse and super coarse particles where the Stoke's approximation is no longer valid, the steady state equation of motion that has to be solved for the determination of the settling velocity is no longer linear, and numerical methods have to be used instead. Drakaki et al. (2022) used a computationnally expensive bisection method. Nevertheless, the inclusion of particles beyond the Stoke's approximation revealed that a reduction of settling velocity around 60-80% is required for their simulation results to agree with airborne and spaceborne measurements.

Mailler et al. (2023c) improved this computational inefficiency by providing a semi-analytical solution to the drag equation based on the Clift and Gauvin (1971) drag coefficient, eliminating the need of numerical iterations required by the numerical solution of this non linear equation and keeping the numerical error compared to Clift and Gauvin (1971) below 2%. Their method improved the computational speed by a factor around 4. The formalism of Mailler et al. (2023c) based on the Clift and Gauvin (1971) drag coefficient is therefore a fast and accurate computational scheme for the study of the settling velocity of spherical particles of all sizes. This formulation has been implemented by the same authors in AerSett v1.0 (Mailler et al., 2023b), a Fortran module designed for inclusion in chemistry-transport models, and already included in Chimere v2023r1 (Menut et al., 2023).

The goal of the current work is to expand this formulation to the case of non spherical solid particles, focusing on prolate spheroids. As in Mailler et al. (2023c), the point of this study is to obtain an explicit and computationally efficient method for the calculation of the settling speed as a function of known properties of the flow and of the particle. This problem is reciprocal of the classical problem in fluid mechanics (calculating the force as a function of the speed). In atmospheric science,

the characteristics of the particle, including the gravity force it is submitted to, are known, while the settling speed is not known *a priori*, making this classical approach impractical for our problem.

To obtain such an explicit expression of the speed as a function of the other parameters, the first point that has to be addressed is the choice of an accurate expression for the drag coefficient in the case of prolate spheroids. In the Stokes regime,
exact analytical solutions similar to the Stokes law for spherical particles (Stokes, 1851) give the values of the drag coefficients in the case of vertically and horizontally orientated prolate spheroids (e.g. Oberbeck, 1876; Jeffery and Filon, 1922; Chwang and Wu, 1975), that can be easily generalized for an arbitrary orientation angle. Additionally, there are higher order expansions that further increase the accuracy of the calculated drag force (Breach, 1961; Chwang and Wu, 1976).

Many efforts have been made in the past for the correction of the drag coefficient expressions for larger particle beyond the
Stokes regime, using different methodologies. In the past literature there are expressions that have been derived using empirical data (e.g. Bagheri and Bonadonna, 2016, 2019; Dioguardi et al., 2018, and references therein), or using Computational Fluid Dynamics (CFD) simulations (e.g. Zastawny et al., 2012; Fröhlich et al., 2020; Sanjeevi et al., 2022, and references therein), or based on theoretical and semi-analytical approximations (e.g. Chwang and Wu, 1976; Mallios et al., 2020, and references therein). It is noted that the correlations derived by empirical data assume mainly random orientation of the particles, while the
correlations based on CFD simulations and the semi analytical approximations take into account the modification of the drag coefficient expression based on the orientation angle of the particle. Indications of preferential orientation of settling prolate spheroids has been established both in theoretical and observational basis (e.g. Klett, 1995; Ulanowski et al., 2007; Mallios et al., 2021).

The choice of an appropriate drag coefficient expression is important because it can alter the physical results. Ginoux (2003)
using a drag coefficient expression by Boothroyd (1971) showed that the terminal velocities of randomly oriented prolate spheroids and of spheres with the same cross section are practically the same, as long as the aspect ratio of the spheroids is less than 5. On the other hand, Huang et al. (2020) using a drag coefficient expression by Bagheri and Bonadonna (2016), concluded that randomly oriented ellipsoids fall around 20% slower than spheres of the same volume, regardless the aspect ratio. Finally, Mallios et al. (2020) using semi analytical expressions for the drag coefficient of prolate spheroids in the case
of vertical and horizontal orientation, derived that horizontally oriented spheroids fall slower than spheres of the same volume, while vertically oriented spheroids fall faster than spheres of the same volume. Moreover, they showed that the difference between the velocities of these two extreme orientation cases can be significant even for small aspect ratio values (around 2).

The goal of this article is to provide explicit and computationally efficient expressions for the calculation of the settling velocity of prolate spheroids, valid for a large range of sizes and aspect ratio. This methodology can be seen as an extension
of the work presented by Mailler et al. (2023c) in the case of spheres. We focus on two available drag coefficient expressions that take into account the orientation of the prolate spheroid and are valid for a wide range of sizes. One is the expression by Mallios et al. (2020) based on theoretical arguments, and the second is an accurate expression by Sanjeevi et al. (2022) derived by heuristic methods based on CFD simulations. We will also describe AerSett v2.0.2, a Fortran module designed to calculate accurately and efficiently the settling speed of prolate particles oriented either horizontally or vertically in the atmosphere.

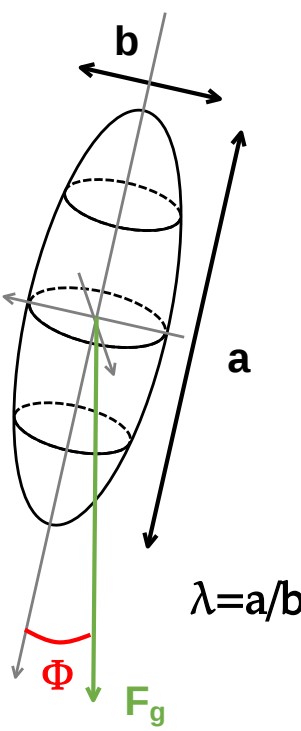

**Figure 1.** Sketch summarizing the main characteristics of the falling spheroid: polar diameter $a$, equatorial diameter $b$, aspect ratio $\lambda = \frac{a}{b}$ and orientation $\phi$ relative to the gravity force $\mathbf{F}_g$

In Section 2 we will expose our new formulation for the expression of the setling speed for prolate spheroids and apply it to the semi-analytical drag formulation of Mallios et al. (2020). In Section 3 we will apply the same method to the accurate drag expressions of Sanjeevi et al. (2022) and examine the differences in comparison to the results in Section 2. In Section 4 we will present the implementation of the method described in Section 2 in AerSett v2.0.2, and we will give our conclusions in Section 5.

## 2 Expressing the settling speed from the parameters of the problem

### 2.1 Description of the problem

We consider a prolate spheroid, with polar diameter $a$, and equatorial diameter $b$ (Fig. 1). By definition, $b < a$ for a prolate spheriod, so that $a$ is sometimes called the major axis, and $b$ the minor axis. The aspect ratio $\lambda$, defined as:

$$\lambda = \frac{a}{b} > 1, \tag{1}$$

is greater than 1 (Fig. 1). Let $e$ be the eccentricity of the spheroid:

$$e = \sqrt{1 - \lambda^{-2}}. \tag{2}$$

we have $e = 0$ for a sphere ($\lambda = 1$) and $0 < e < 1$ for a prolate spheroid ($\lambda > 1$). The volume of this spheroid is:

$$V = \frac{\pi a b^2}{6}. \tag{3}$$

We define the volume-equivalent diameter of this spheroid (also known as mass-equivalent diameter) as the diameter of the sphere with equal volume:

$$d_{eq} = \left( \frac{6V}{\pi} \right)^{\frac{1}{3}} = a \lambda^{-\frac{2}{3}} \tag{4}$$

Finally, we introduce $\phi$ the angle of the polar axis of the spheroid relative to the vertical direction (defined as the direction of the gravity force vector, see Fig. 1).

Let us now suppose that this spheroid is a material particle with density $\rho_p$ falling under the effect of gravity in a fluid, and that this particle is oriented either vertically ($\phi = 0$) or horizontally ($\phi = \pi/2$). In these configurations, no lift force and no torque are exerted by the fluid on the particle, so that the particle can fall vertically, with speed $\mathbf{u}$ colinear to the acceleration of gravity $\mathbf{g}$.

## 2.2 Method for the calculation of the settling-speed in the continuous case

The flow around the settling prolate spheroid (or any object in general) is characterized by the Reynolds number $Re = \frac{\rho U L}{\mu}$ where $\mu$ is the dynamic viscosity of the fluid, $\rho$ its mass density, $U$ the speed of the fluid relative to the object, and $L$ a characteristic length. The magnitude of the drag force exerted upon the object is typically expressed as $F_D = \frac{1}{2} \rho \mathcal{A}_p C_D \left( Re; \phi \right) U^2$, where $\mathcal{A}_p$ is the cross-flow projected area. This formalism, the most common in aerodynamical studies, is the one used in Mallios et al. (2020), but has the inconvenience that both the cross-flow projected area $\mathcal{A}_p$ and the drag coefficient $C_D$ depend on the orientation of the object relative to the flow. In Sanjeevi et al. (2022), the Reynolds number $Re$ and the drag coefficient $C_D$ are defined in terms of the volume-equivalent diameter $d_{eq}$ as follows:

$$Re = \frac{|\mathbf{u}| \, d_{eq} \rho}{\mu} \tag{5}$$

$$F_D = C_D \left( Re, \phi \right) \times \frac{1}{2} \rho |\mathbf{u}|^2 \frac{\pi}{4} d_{eq}^2, \tag{6}$$

where the drag coefficient $C_D \left( Re, \phi \right)$ depends only on the particle's shape and orientation, and on the Reynolds number.

For spheres, Stokes (1851) has proved that for $Re \ll 1$ we have $C_D \simeq \frac{24}{Re}$. This formulation can be extended to prolate spheroids, but with a different multiplicative constant:

$$C_D \simeq \frac{A^{\lambda, \phi}}{Re} \text{ for } Re \ll 1. \tag{7}$$

where the expressions for $A^{\lambda, \phi}$ $\left( \phi \in \left\{ 0, \frac{\pi}{2} \right\} \right)$, also known as shape factors, are derived by the exact analytical solution of the Navier-Stokes equation coupled with the continuity equation for the creeping flow of an incompressible viscous fluid past a

prolate spheroid (Oberbeck, 1876; Jeffery and Filon, 1922; Chwang and Wu, 1975):

$$A^{\lambda,\phi=0} = 64\lambda^{2/3} \times e^3 \left[-2e + \left(1+e^2\right)\log\left(\frac{1+e}{1-e}\right)\right]^{-1} \tag{8}$$

$$A^{\lambda,\phi=\frac{\pi}{2}} = 64\lambda^{2/3} \times 2e^3 \left[2e + \left(3e^2-1\right)\log\left(\frac{1+e}{1-e}\right)\right]^{-1}. \tag{9}$$

It is noted that, $\lambda \to 1$, both $A^{\lambda,\phi=0}$ and $A^{\lambda,\phi=\frac{\pi}{2}}$ tend to 24, transforming Eq. 7 to the well-known expression for the drag coefficient of a sphere.

The above drag coefficient expression can be generalized for cases other than creeping flow after multiplication by a correction function $\mathcal{D}(Re)$:

$$C_D(Re) = \frac{A^{\lambda,\phi}}{Re}\mathcal{D}(Re), \text{ with } \lim_{Re\to 0^+}\mathcal{D}(Re) = 1, \tag{10}$$

where function $\mathcal{D}$ can be named as *drag function*. There is no exact analytical expression of the drag function for the whole range of Reynolds numbers. Mallios et al. (2020) give an expression of this function using theoretical arguments to extend the Clift and Gauvin (1971) empirical formula to prolate spheroids, while Sanjeevi et al. (2022) provides another estimate of $\mathcal{D}$ based on numerical CFD simulations.

The settling velocity $v_\infty$ can be calculated by the steady-state Newton's law, where the drag force and the buoyancy force counterbalance the gravitational force, leading to a net force equal to zero:

$$v_\infty = \frac{4}{3}\frac{(\rho_p - \rho)\,g d_{eq}^2}{A^{\lambda,\phi}\mu\mathcal{D}(Re)} \tag{11}$$

$$= \frac{U^{\lambda,\phi}}{\mathcal{D}(Re)}, \tag{12}$$

where $U^{\lambda,\phi} = \frac{4}{3}\frac{(\rho_p-\rho)g d_{eq}^2}{A^{\lambda,\phi}\mu}$ is the settling velocity of a prolate spheroid with aspect ratio $\lambda$ and orientation angle $\phi$ supposing that the Stokes law is verified exactly. On the other hand, $v_\infty$ is the settling speed of the same prolate spheroid taking into account the deviations from the Stokes law, reflected in the $\mathcal{D}(Re)$ drag function. In particular, the Stokes settling speed for the sphere is:

$$U = \frac{(\rho_p - \rho)\,g d_{eq}^2}{18\mu}. \tag{13}$$

When the particle reaches the terminal settling speed $v_\infty$, we have:

$$Re = \frac{v_\infty d_{eq}\rho}{\mu}. \tag{14}$$

We introduce the Archimedes number $Ar$ (called "virtual Reynolds number" in Mailler et al. (2023c)). The Archimedes number is equal to the Reynolds number of a sphere having the same volume as the prolate spheroid and obeying the Stokes law (Eq. 13):

$$R = \frac{U d_{eq}\rho}{\mu} = \frac{(\rho_p - \rho)\rho g d_{eq}^3}{18\mu}, \tag{15}$$

Equation 12 becomes:

$$\frac{v_\infty}{U^{\lambda,\phi}} = [\mathcal{D}(Re)]^{-1} = \left[\mathcal{D}\left(\frac{v_\infty}{U^{\lambda,\phi}}\frac{24}{A^{\lambda,\phi}}Ar\right)\right]^{-1} \tag{16}$$

We now introduce the speed function $\mathcal{S}$ as:

$$\mathcal{S} = \frac{v_\infty}{U^{\lambda,\phi}}, \tag{17}$$

so that $\mathcal{S}$ is the solution to the fixed-point equation:

$$\mathcal{S} = \left(\mathcal{D}\left(\frac{24}{A^{\lambda,\phi}}Ar \cdot \mathcal{S}\right)\right)^{-1}. \tag{18}$$

As we will see later, solving Eq. 18 permits to express $\mathcal{S}$ as a function of the parameters of the problem, in particular of the virtual Reynolds number $R$. Once this is done, the settling speed can be found as:

$$v_\infty = \mathcal{S}(Ar) \cdot U^{\lambda,\phi} \tag{19}$$

## 2.3 Inclusion of the slip-correction factor

For the slip-correction factor, we adopt the formulation of Fan and Ahmadi (2000), based on the Adjusted-Sphere-Approximation (ASA) introduced by Dahneke (1973). These authors give the following expressions for the slip-correction factors:

$$C_c^\phi = 1 + \mathrm{Kn}^\phi\left[1.257 + 0.4\exp\left(\frac{-1.1}{\mathrm{Kn}^\phi}\right)\right], \ \mathrm{Kn}^\phi = \frac{\ell}{r^\phi}\ \left(\phi \in \left\{0; \frac{\pi}{2}\right\}\right), \tag{20}$$

where $\ell = \sqrt{\frac{\pi}{8}}\frac{\mu}{0.4987445}\frac{1}{\sqrt{\rho P}}$ is the mean-free path of air molecules, $P$ being the atmospheric pressure (Jennings, 1988).

The radius of the adjusted sphere moving in the polar-axis direction, $r^{\phi=0}$, and that of the adjusted sphere moving in the *equatorial* direction, $r^{\phi=\pi/2}$, are given by:

$$r^{\phi=0} = \frac{1.657\,a}{8\,(\lambda^2-1)}\left[\frac{2\lambda^2-1}{\sqrt{\lambda^2-1}}\ln\left(\lambda+\sqrt{\lambda^2-1}\right)+\lambda\right]\left\{2E_p f + \frac{G_p}{e^2}\left[e^2\,(4-2f)-4+\left(3-\frac{\pi}{2\lambda^2}\right)f\right]\right\} \tag{21}$$

$$r^{\phi=\frac{\pi}{2}} = \frac{1.657\,a}{16\,(\lambda^2-1)}\left[\frac{2\lambda^2-3}{\sqrt{\lambda^2-1}}\ln\left(\lambda+\sqrt{\lambda^2-1}\right)+\lambda\right]\left\{E_p\left[4+\left(\frac{\pi}{2}-1\right)f\right]+\frac{G_p}{e^2}\left(2+\frac{4e^2+\pi-6}{4}f\right)\right\}. \tag{22}$$

In Equations 21-22, $e$ is the spheroid's eccentricity as defined in Eq. 2, $E_p = \frac{\sin^{-1}e}{e}$, and $G_p = \frac{1}{\lambda} - E_p$. Following Mallios et al. (2020), we adopt the value $f = 0.9113$ for the "momentum accomodation coefficient".

With the inclusion of the slip-correction factor, Eq. 11 is modified as follows:

$$\widetilde{v}_\infty = \frac{4}{3}\frac{C_c^\phi\,(\rho_p - \rho)\,g d_{eq}^2}{A^{\lambda,\phi}\mu\mathcal{D}(Re)}. \tag{23}$$

Hereinafter, the variables including the effect of free-slip correction ($C_c^\phi$ in Eq. 23) will be indicated by a tilde ($\widetilde{\cdot}$). As such, we introduce the Stokes settling speed including the slip-correction term, $\widetilde{U}^\phi$, as:

$$\widetilde{U}^{\lambda,\phi} = C_c^\phi U^{\lambda,\phi} \tag{24}$$

With this, in a similar way as in Mailler et al. (2023c), we obtain:

$$\widetilde{v}_\infty = \mathcal{S}\left(\widetilde{Ar}\right) \cdot \widetilde{U}^{\lambda,\phi}, \tag{25}$$

with

$$\widetilde{Ar} = C_c^\phi \frac{d_{eq}^3 \rho\left(\rho_p - \rho\right) g}{18\mu^2}. \tag{26}$$

Function $\mathcal{S}$ in Eq. 25 is the same as in the case without slip-correction, defined by Eq. 18.

## 2.4 Application to the Mallios et al. (2020) drag formulation

At this point, we would like to mention that there are two typos in the equations of Mallios et al. (2020):

1. in the drag coefficient expression for the horizontally oriented particles (their Eqs. 22 and 41), the expressions should be multiplied with the aspect ratio.

2. in the projected area of the horizontally oriented spheroid (their Eq. 45), the expression should be divided by the aspect ratio.

These two modifications cancel each other, since the drag coefficient is multiplied with the projected area for the calculation of the drag force, so that the equations governing the settling speed for horizontally oriented spheroids (their Eqs. 50 and 52) are eventually correct. This means that the conclusions of Mallios et al. (2020) are not affected by these typos, and that the Corrigendum that was published later by the authors addressing only one of the two typos should not be taken into account, since it would lead to erroneous results.

The drag coefficient formulation of Mallios et al. (2020) converted to our notations is:

$$C_{D,M20}^{\lambda,\phi} = \frac{A^{\lambda,\phi}}{Re} F_{cg}\left(\frac{A^{\lambda,\phi}}{24} Re\right), \tag{27}$$

$$\mathcal{D}\left(Re\right) = F_{cg}\left(\frac{A^{\lambda,\phi}}{24} Re\right) \tag{28}$$

where

$$F_{cg}\left(x\right) = 1 + 0.15x^{0.687} + \frac{0.42x}{24}\left(1 + \frac{42500}{x^{1.16}}\right)^{-1}. \tag{29}$$

Figures 2 and 3 compare the drag formulation of Mallios et al. (2020) (Eq. 27) to that of Sanjeevi et al. (2022) (see Eq. 35 below). Fig. 2a shows that, for spherical particles, both drag formulations give extremely similar results at least up to Re=300. For prolate spheroidal particles (Fig. 2b,c,d), we observe that the drag formulation of Mallios et al. (2020) is comparable to that of Sanjeevi et al. (2022) for horizontally oriented particles up to Re=300. On the other hand, for vertically oriented particles, we see that substantial differences arise between both formulations, in particular for particles that have a strong

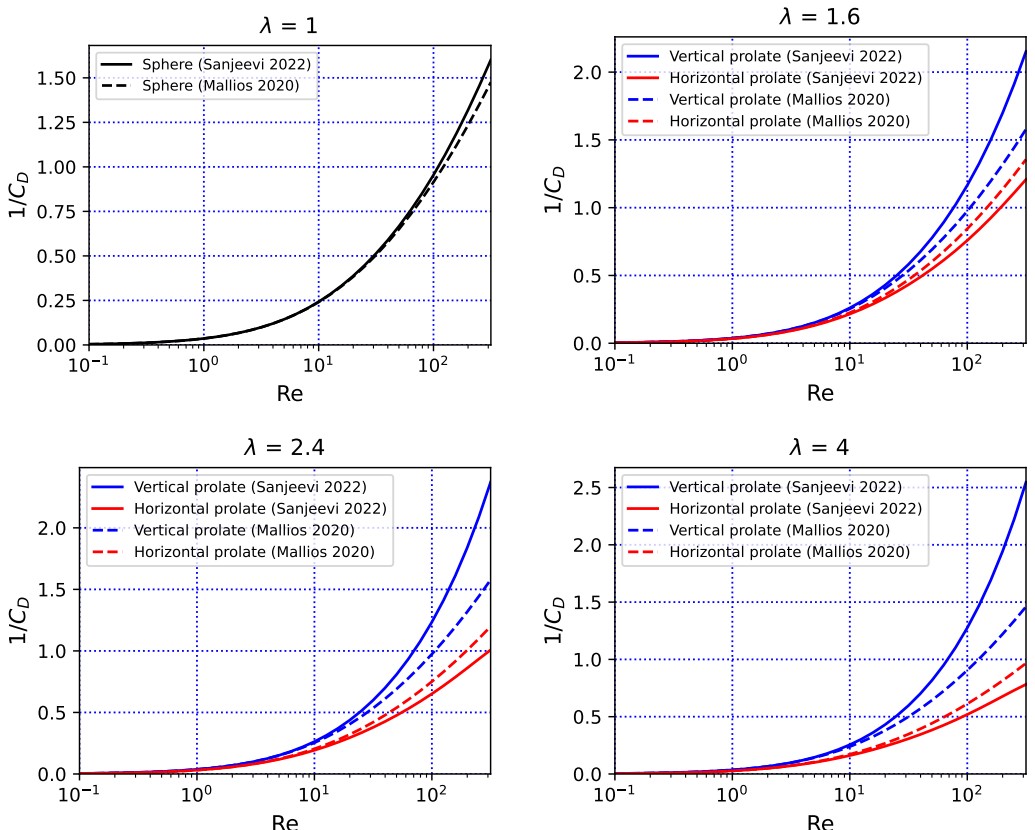

**Figure 2.** $1/C_D$ as a function of Re for (a) $\lambda = 1$; (b) $\lambda = 1.6$; (c) $\lambda = 2.4$; and (d) $\lambda = 4$. The plots represent $1/C_D$ rather than $C_D$ to avoid hiding the differences that occur for high Re.

aspect ratio. These differences are not problematic for our application because, as we will discuss below, high values of Re are reached only by the coarsest atmospheric particles, and as shown in, *e.g.*, Mallios et al. (2021), such particles tend to fall with a horizontal orientation. Figure 3 confirms the good agreement between the formulations of Mallios et al. (2020) and Sanjeevi et al. (2022) for both vertical and horizontal orientations at low Reynolds number (Fig. 3a-b). At higher values of Re (Fig. 3c-d), a reasonable degree of agreement persists in the horizontal orientation, but substantial differences arise at high values of Re in the vertical orientation. Generally speaking, for Re$\geq$100, the drag coefficient as calculated from the Mallios et al. (2020) is slightly weaker than the estimate of Sanjeevi et al. (2022) when the particle is oriented horizontally, but much stronger than the estimate of Sanjeevi et al. (2022) when the particle is oriented vertically. As we will see below, this will be reflected into stronger discrepancies between both methods for vertically oriented particles than for horizontally oriented particles.

Eq. 18 with $C_D$ as expressed in 27 yields:

$$\mathcal{S} = \left(F_{cg}\left(Ar \cdot \mathcal{S}\right)\right)^{-1}. \tag{30}$$

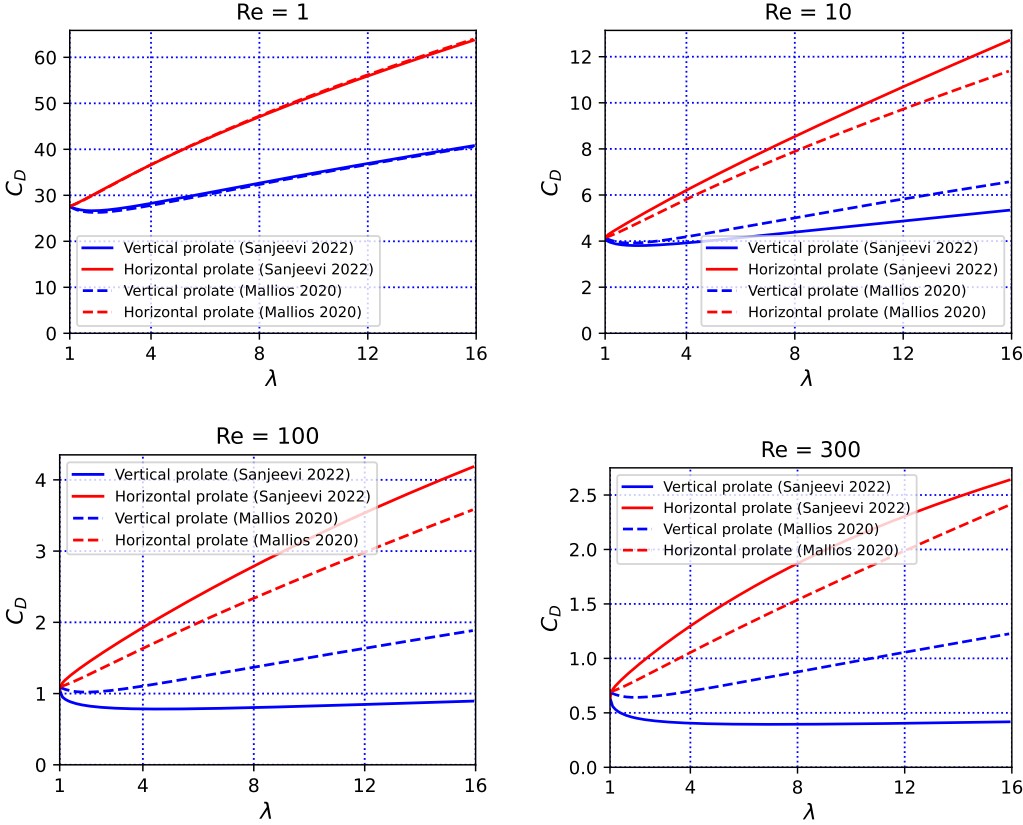

**Figure 3.** $C_D$ as a function of $\lambda$ for (a) Re $= 1$; (b) Re $= 10$; (c) Re $= 100$; and (d) Re $= 300$.

An equivalent fixed-point equation has been solved in Mailler et al. (2023c) (their Eqs. 13 and 16), yielding the following approximated expression for $\mathcal{S}(Ar)$:

$$\mathcal{S}(Ar) = 1 - \left[1 + \left(\frac{Ar}{4.880}\right)^{-0.4335}\right]^{-1.905}. \tag{31}$$

As discussed in Mailler et al. (2023c), using this explicit formula instead of numerically resolving Eq. 30 induces a loss of less than 2.5% in accuracy for $Re < 1000$, which is not critical since, the uncertainty of the Clift-Gauvin formula itself (and of other comparable drag-coefficient formulations) is around 7% when compared to experimental measurements (Goossens, 2019).

This expression of $\mathcal{S}$ yields the following expression for $v_\infty$ in the absence of slip-correction:

$$v_\infty^{\lambda,\phi} = \frac{4(\rho_p - \rho)gd_{eq}^2}{3\mu A^{\lambda,\phi}} \mathcal{S}\left(\frac{d_{eq}^3\rho(\rho_p - \rho)g}{18\mu^2}\right), \text{ for } \phi \in \left\{0; \frac{\pi}{2}\right\}, \tag{32}$$

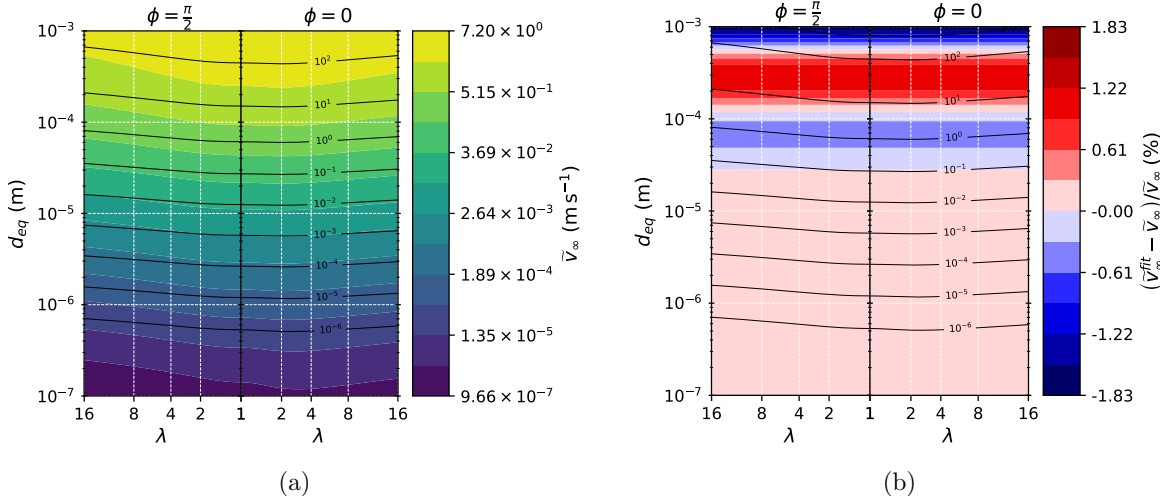

**Figure 4.** (a) $\widetilde{v}_\infty^{\lambda,\phi}$ as a function of $d_{eq}$ and $\lambda$ for $\phi = \frac{\pi}{2}$ (left panel) and $\phi = 0$ (right panel), using Eq. 31 to estimate $\mathcal{S}$; and (b) error (in %) commited by using explicit expression Eq. 31 instead of solving numerically Eq. 30. Contours in (a) and (b) represent the Reynolds number. This figure is produced for standard atmospheric conditions ($p = 101325\,\mathrm{Pa}$, $T = 298.15\,\mathrm{K}$).

and in the presence of slip-correction:

$$\widetilde{v}_\infty^{\lambda,\phi} = \frac{4 C_c^\phi (\rho_p - \rho)\, g d_{eq}^2}{3\mu A^{\lambda,\phi}} \mathcal{S}\left( \frac{C_c^\phi d_{eq}^3 \rho (\rho_p - \rho)\, g}{18\mu^2} \right), \text{ for } \phi \in \left\{ 0; \frac{\pi}{2} \right\}. \tag{33}$$

Fig. 4a shows the evaluation of $\widetilde{v}_\infty^{\lambda,\phi}$ from Eq. 33, and Fig. 4b shows the numerical error due to using Eq. 33 rather than
numerically solving the fixed-point equation 30. Both panels of Fig. 4 as well as all the subsequent figures in the study have been produced using standard atmospheric consitions for air ($p = 101325\,\mathrm{Pa}$ and $T = 298.15\,\mathrm{K}$). Dynamic viscosity $\mu$ has been calculated following the US Standard Atmosphere (NOAA/NASA/USAF, 1976):

$$\mu = \frac{\beta T^{\frac{3}{2}}}{T + S}, \tag{34}$$

where $\beta = 1.458 \times 10^{-6}\,\mathrm{kg\,s^{-1}\,m^{-1}\,K^{-\frac{1}{2}}}$ and $S = 110.4\,\mathrm{K}$. In these conditions of temperature and pressure and with the molar
mass of dry air $M_a = 28.9644 \times 10^{-3}\,\mathrm{kg\,mol^{-1}}$ (also from the US Standard Atmosphere), the density of air is $\rho = 1.18\,\mathrm{kg\,m^{-3}}$.

Figure 4a shows that $\widetilde{v}_\infty^{\lambda,\phi}$ depends essentially on the particle diameter $d_{eq}$, but also on aspect ratio $\lambda$ as expected. A closer look at Fig. 4a reveals that, for vertically oriented particles, $\widetilde{v}_\infty^{\lambda,\phi}$ is at first increasing with increased aspect ratio, but from $\lambda \simeq 4$ this evolution is reversed. Sanjeevi et al. (2022) explain this feature as a tradeoff between the pressure drag and the viscous drag. While the pressure drag continuously decreases with particle elongation (for vertical particles), the viscous drag tends
to increase due to the higher surface-area of the particle with increasing $\lambda$. Fig. 4b shows that the error due to using explicit expression 31 induces a difference less than 2% for all equivalent diameters up to $10^3\,\mu\mathrm{m}$.

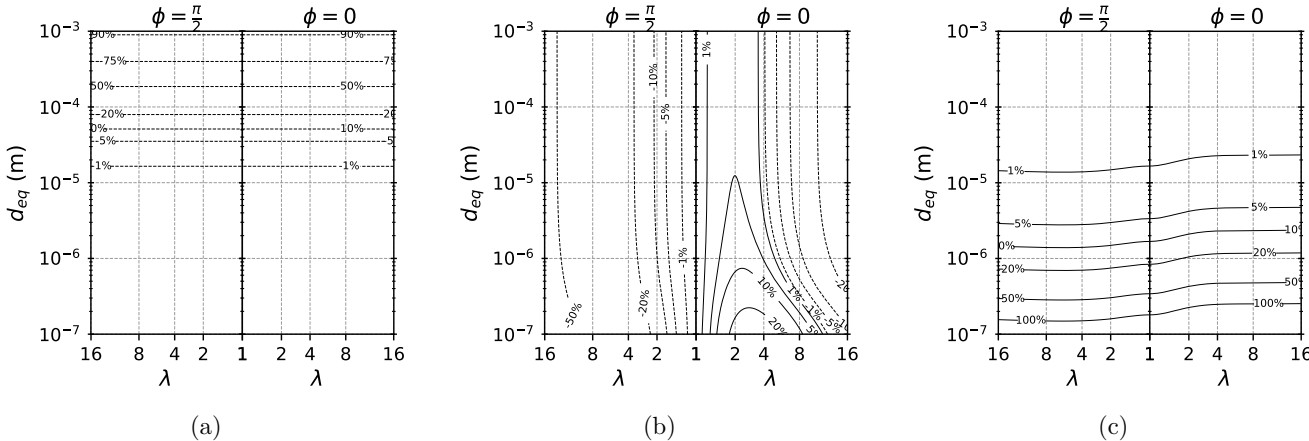

**Figure 5.** (a) Large-particle correction $\frac{\widetilde{v}_\infty - \widetilde{U}^{\lambda,\phi}}{\widetilde{U}^{\lambda,\phi}}$; (b) eccentricity correction $\frac{\widetilde{v}_\infty(\lambda;d_{eq}) - \widetilde{v}_\infty(\lambda=0;d_{eq})}{\widetilde{v}_\infty(\lambda=0;d_{eq})}$; and (c) slip-correction $\frac{\widetilde{v}_\infty(\lambda;d_{eq}) - v_\infty(\lambda;d_{eq})}{v_\infty(\lambda;d_{eq})}$. The three panels are in % (contours). The figure is produced for standard atmospheric conditions ($p = 101325\,\text{hPa}$, $T = 298.15\,\text{K}$)

## 2.5 Sensitivity of the settling speed on large-particle corrections, eccentricity and slip-correction

Figure 5 shows the effects of large-particle correction (Fig. 5a), eccentricity correction (Fig. 5b) and slip correction (Fig. 5c) on the settling speed, showing that large-particle correction begins to be significant ($< -5\%$) for particles with $d_{eq} > 30\,\mu\text{m}$. On the contrary, slip correction is significant ($> 5\%$) only for particles with $d_{eq} < 3 - 5\,\mu\text{m}$ (depending on particle eccentricity). For lower pressure values ($p \simeq 200\,\text{hPa}$) representative of the higher troposphere or lower stratosphere, slip correction increases due to the longer mean-free path for air particles in thinner air. At these altitudes, slip-correction reaches $5\%$ for particles with $d_{eq} < 8 - 15\,\mu\text{m}$ (not shown), while large-particle corrections also reaches $-5\%$ for particles with $d_{eq} > 30\,\mu\text{m}$ (not shown). Total correction to the Stokes velocity of the volume-equivalent sphere (including the effect of eccentricity, slip correction and large-particle correction) is shown on Fig. 6, revealing that the magnitude of the effect behaves differently for horizontal and vertical particles. Horizontal particles always fall more slowly than their volume-equivalent sphere. For aspect ratio $\lambda > 2$, the difference is around $10\%$, showing a possible interest of this difference from a modelling point of view. For vertically-oriented particles, and except for the smaller ones (influenced by slip-correction factors), the difference in $\widetilde{v}_\infty$ due to particle eccentricity does not exceed $\pm 10\%$ until $\lambda \simeq 7$. Figure 6, including both eccentricity and large-particle effects, show the difference between the model we propose here (Eq. 33) and the expression typically used in chemistry-transport models, assuming particles to be spherical and not taking into account large-particle correction. Consistently with Figs. 5a-b, it shows that these effects need to be into account when $\lambda > 2$ and/or $d_{eq} > 50\,\mu\text{m}$, and that the eccentricity effect is much stronger of horizontally-oriented particles than for vertically-oriented particles.

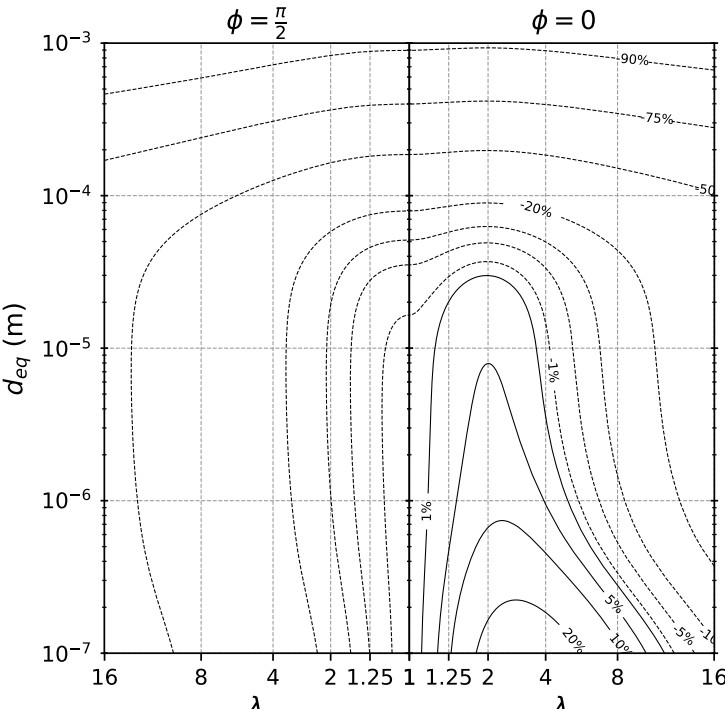

**Figure 6.** Total correction to the slip-corrected Stokes speed for a volume-equivalent sphere, $\frac{\widetilde{v}_\infty - \widetilde{U}^{\lambda,\phi}}{\widetilde{U}^{\lambda,\phi}}$. This figure is produced for standard atmospheric conditions ($p = 101325\,\text{Pa}$, $T = 298.15\,\text{K}$).

## 3  Comparison with the Sanjeevi et al. (2022) drag formulation

### 3.1  Expression of the settling speed from the Sanjeevi et al. (2022) drag formulation

Sanjeevi et al. (2022) authors suggest the following form:

$$C_{D,S22}^{\lambda,\phi} = \left( \frac{A^{\lambda,\phi}}{Re} + \frac{a_2^{\lambda,\phi}}{Re^{a_3^{\lambda,\phi}}} \right) e^{-a_4^{\lambda,\phi} Re} + a_5^{\lambda,\phi} \left( 1 - e^{-a_4^{\lambda,\phi} Re} \right), \text{ for } \phi \in \left\{ 0; \frac{\pi}{2} \right\}. \tag{35}$$

While $a_3^{\lambda,\phi} < 1$, these equations guarantee that $C_D^{\lambda,\phi} = \frac{A^{\lambda,\Phi}}{Re}$ is the dominant term for $Re \ll 1$. Coefficients $a_2^{\lambda,\phi}$ to $a_5^{\lambda,\phi}$ have been determined empirically by Sanjeevi et al. (2022), and these authors give the necessary expressions, dependant on $\lambda$, in their Eqs. 11-12 and Table 2. Therefore, combining Equation 35 with the expressions of the $a_i$ coefficients, the above elements permit to completely express the drag coefficient $C_D$ as a function of $Re$, the aspect ratio $\lambda$ and the attack angle $\phi$ for all the falling spheroids.

| $i$ | $A_i$ | $B_i^0$ | $C_i^0$ | $D_i^0$ | $E_i^0$ | $F_i^0$ | $B_i^{\pi/2}$ | $C_i^{\pi/2}$ | $D_i^{\pi/2}$ | $E_i^{\pi/2}$ | $F_i^{\pi/2}$ |
|---|---|---|---|---|---|---|---|---|---|---|---|
| 1 | 0.4124 | 0.05654 | -0.05935 | -0.1139 | 0.05216 | $4.162\times10^{-5}$ | 0.05592 | -0.0595 | -0.01901 | -0.02243 | $3.659\times10^{-5}$ |
| 2 | 0.8466 | 6.836 | -6.608 | -7.255 | 2.17 | -0.004809 | -1.228 | 1.118 | 1.15 | -1.009 | -0.005533 |
| 3 | -2.389 | 4.377 | -4.291 | -4.357 | 1.113 | -0.002111 | 0.2318 | -0.4136 | 0.1078 | -0.9109 | $-2.913\times10^{-5}$ |

**Table 1.** Values of the coefficients in Eqs. 38-39

The expression of $\mathcal{D}$ (Eq. 10) from the expressions of Sanjeevi et al. (2022) (Eq. 35) is as follows, with either $\phi = 0$ or $\phi = \frac{\pi}{2}$:

$$\mathcal{D} = \left(1 + \frac{a_2^{\lambda,\phi}}{A^{\lambda,\phi}} \left(Re\right)^{1-a_3^{\lambda,\phi}}\right) e^{-a_4^{\lambda,\phi} Re} + \frac{a_5^{\lambda,\phi}}{A^{\lambda,\phi}} Re \left(1 - e^{-a_4^{\lambda,\phi} Re}\right). \tag{36}$$

With this expression of $\mathcal{D}$, it is possible to solve numerically Eq. 18 to obtain the values of $\mathcal{S}$ as a function of $\lambda$, $\phi \in \{0; \pi/2\}$ and $R$. In Mailler et al. (2023c), we have see that, for spherical particles, it is possible to express $\mathcal{S}$ as a function of $R$ with a high degree of accuracy as:

$$\mathcal{S}\left(Ar\right) \simeq - \left(1 + e^{-c_1^\phi(\lambda)\left(\ln Ar - c_2^\phi(\lambda)\right)}\right)^{c_3^\phi(\lambda)}, \tag{37}$$

We have found that approximated expressions of the following form hold for the $c_i^\phi$ coefficients:

$$c_i^\phi = A_i + B_i^\phi e\lambda \quad + C_i^\phi \left(\lambda - 1\right) \quad + D_i^\phi e \quad + E_i^\phi \frac{e}{\lambda} \quad + F_i^\phi \left(e\lambda\right)^2 \quad \text{for } c_{1,2,3}^0 \text{ and for } c_{1,2}^{\pi/2} \tag{38}$$

$$c_3^{\pi/2} = A_3 + B_3^{\pi/2} e\lambda + C_3^{\pi/2} \left(\lambda - 1\right) + D_3^{\pi/2} e + E_3^{\pi/2} \frac{e}{\lambda} + F_3^{\pi/2} \left(e\lambda\right)^4, \tag{39}$$

The values of $A_i$, $B_i^\phi$, $C_i^\phi$, $D_i^\phi$, $E_i^\phi$ and $C_i^\phi$ for $i \in \{1; 2; 3\}$ and $\phi \in \{0; \pi/2\}$ are given in Table. 1.

Fig. 7a shows the evaluation of $\widetilde{v}_\infty^{\lambda,\phi}$ using $\mathcal{S}$ from Eq. 37, and Fig. 7b shows the numerical error due to using Eq. 37 rather than numerically solving Eq. 18. Figure. 7a shows a behaviour very similar to the Mallios et al. (2020) formulation (a more detailed comparison of the results will be provided below). Fig. 7b shows that the error attributable to the numerical fit of $\mathcal{S}$ by Eq. 37 is very small, less than $2.6\%$ in all the represented domain. Therefore, there is no inconvenience in using this approximated expression for $\mathcal{S}$.

### 3.2 Comparison of the speed expressions from Mallios et al. (2020) and Sanjeevi et al. (2022)

Figure 8 shows the relative difference between the estimation of $\widetilde{v}_\infty$ from the Sanjeevi et al. (2022) drag formulation and from the Mallios et al. (2020) drag formulation. Up to $d_{eq} \simeq 10^{-4}\,\mathrm{m}$, the difference between both formulations is below or around $2\%$. Differences are more substantial for $d_{eq} > 10^{-4}\,\mathrm{m}$ and for the vertically oriented particles. For horizontally oriented particles, differences stay below or around $10\%$ up to $d_{eq} = 10^{-3}\,\mathrm{m}$, which is close to the uncertainty range of both the Clift and Gauvin (1971) drag formulation (see Goossens (2019)) and the Sanjeevi et al. (2022) formulation. This is particularly

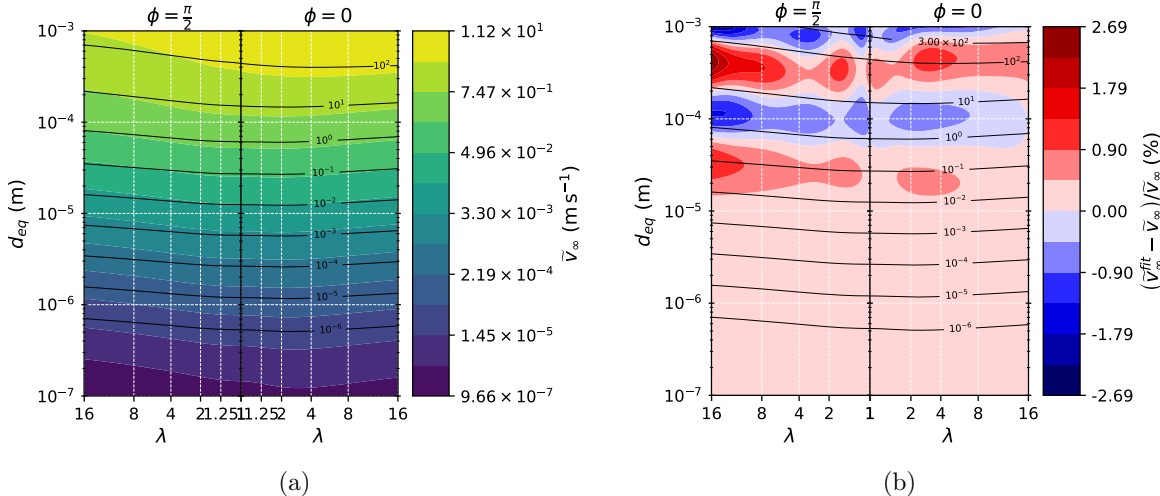

**Figure 7.** (a) $\widetilde{v}_\infty^{\lambda,\phi}$ as a function of $d_{eq}$ and $\lambda$ for $\phi = \frac{\pi}{2}$ (left panel) and $\phi = 0$ (right panel) using Eq. 37; and (b) error in % committed by using explicit expression Eq. 37 instead of solving numerically Eq. 18. Contours in (a) and (b) represent the Reynolds number. This figure is produced for standard atmospheric conditions ($p = 101325\,\mathrm{Pa}$, $T = 298.15\,\mathrm{K}$).

interesting, since, as shown by Mallios et al. (2021), for reasons of stability, the large and elongated particles with diameter $> 10^{-4}\,\mathrm{m}$ tend to be aligned horizontally. On the contrary, for vertically falling particles, error builds up rapidly, and is in excess of 50% for the biggest and most elongated particles. This tends to show that the eccuracy of the Mallios et al. (2020) model is excellent (close to 2%) for all particle diameters with $d_{eq} < 10^{-4}\,\mathrm{m}$, and good (close to 10%) for all the horizontally oriented particles with $d_{eq} < 10^{-3}\,\mathrm{m}$. All in all, if we suppose that the Sanjeevi et al. (2022) is valid in the range claimed

by these authors ($\lambda < 16$ and $Re < 2000$), our results shows that the accuracy of the theoretical formulation of Mallios et al. (2020), and its application to an explicit expression of $\widetilde{v}_\infty$ (Eq.33) is suitable for the use in atmospheric sciences, for all the atmospheric aerosol which can be reasonably assumed to have a spheric or prolate spheroidal shape.

## 4   Implementation in AerSett v2.0.2

Equation 33 along with Eqs. 8-9 to express $A^{\lambda,\phi}$, Eqs. 20-21-22 to express $C_c^\phi$ and Eq. 31 giving the expression of function $\mathcal{S}$
gives the expression of the settling speed of a falling particle as a function of:

- $d_{eq}$, the mass-equivalent diameter of the particle

- $\rho_p$, density of the particle

- $\rho_a$, density of air

- $\mu$, dynamic viscosity of are

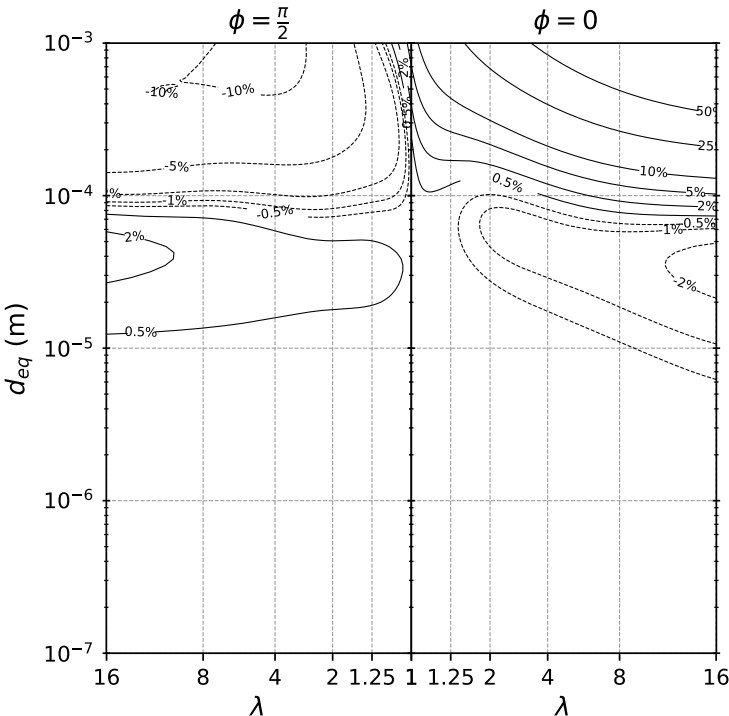

**Figure 8.** Relative difference ( $\frac{\tilde{v}_\infty^{Sj22} - \tilde{v}_\infty^{M20}}{\tilde{v}_\infty^{M20}}$ ) in % between the settling speed estimated from the Sanjeevi et al. (2022) formulation (Eq. 37) and from the Mallios et al. (2020) formulation (Eq. 31). This figure is produced for standard atmospheric conditions ($p = 101325\,\text{Pa}$, $T = 298.15\,\text{K}$).

- – $\ell$, the mean-free path of air molecules

- – $\lambda$, the aspect ratio of the particle

- – $\phi \in \{0°; 90°\}$, the attack angle of the particle

Eq. 33 extends the method exposed for spherical perticles in Mailler et al. (2023c) to the calculation of the settling speed of prolate spheroidal aerosols. This permits to generalize the AerSett module to the calculation of the settling speed of prolate spheroidal aerosol. This section presents this implementation, and qualifies the results of AerSett 2.0.2 in terms of accuracy and numerical efficiency (Table 2).

The U.S Standard Atmosphere (NOAA/NASA/USAF, 1976) has been used as a typical profile for pressure and density. The mass-equivalent diameters of the particles span the range from $10^{-7}\,\text{m}$ to $10^{-3}\,\text{m}$, which has been divided into four intervals (column 1 of Table 2). The mean calculation times for each diameter range are evaluated by calling the calculation routine $10^7$ times on a random sample of $10^7$ triplets of diameter, aspect ratio (between 1 and 16) and altitude (between $1\,\text{m.a.s.l}$ and $12\,000\,\text{m.a.s.l}$) in the case of spheres and prolate spheroids (columns 2-4 of Table 2).

| diameter range (m) | Spherical particle | Calculation times in nanoseconds | | | | | | Maximal relative error between LUT and explicit calculation | |
| --- | --- | --- | --- | --- | --- | --- | --- | --- | --- |
| | | Prolate particle Aersett, no LUT | | Prolate particle Aersett, LUT | | Prolate particle Bisection, LUT | | | |
| | | $\phi = 0°$ | $\phi = 90°$ | $\phi = 0°$ | $\phi = 90°$ | $\phi = 0°$ | $\phi = 90°$ | $\phi = 0°$ | $\phi = 90°$ |
| $10^{-7}$–$10^{-6}$ | 8.8 ns | 76 ns | 76 ns | 24 ns | 24 ns | 24 ns | 24 ns | 0.54% | 0.17% |
| $10^{-6}$–$10^{-5}$ | 8.8 ns | 77 ns | 77 ns | 24 ns | 25 ns | 24 ns | 25 ns | 0.25% | 0.05% |
| $10^{-5}$–$10^{-4}$ | 34 ns | 115 ns | 112 ns | 57 ns | 53 ns | 178 ns | 175 ns | 0.11% | 0.05% |
| $10^{-4}$–$10^{-3}$ | 33 ns | 113 ns | 112 ns | 51 ns | 51 ns | 291 ns | 291 ns | 0.09% | 0.05% |

**Table 2.** Average calculation times to obtain the settling speed of spherical and prolate particles using AerSett (columns 2-6) and using a bisection method (columns 7-8) as a function of the range of mass-equivalent diameter (column 1). Columns 9-10 give the percentage error due to the use of lookup-tables (LUT) for Eqs. 21-22 and 8-9 instead of the formal calculations.
The tests has been performed on a laptop with an Intel Core i7-1165G7 CPU.

To speed up the calculations, the values of $\frac{r^\phi}{a}$ (Eqs. 21-22) and $A^{\lambda,\phi}$ (Eqs. 8-9) for $\phi \in \{0°; 90°\}$ and for $1 < \lambda < 16$ with a step of 0.01 on $\lambda$ are calculated once and for all in the initialization phase and stored in arrays to be used at each call of the calculation routine. This initialization phase takes less than 1 ms on a laptop and needs to be performed only once. The change in performance due to this precalculation of some parameters can be seen in columns 5-6 of Table 2. Finally, Columns 9-10 indicate the percentage error due to the use of lookup-tables for Eqs. 8-9 and 21-22 instead of the formal calculations.

Table 2 shows that the computation time for the settling speed of a prolate particle with AerSett is longer than for a spherical particle, even when a lookup table is used instead of Eqs. 8-9-21-22. However, the use of the LUT strongly reduces this additional cost. Once the LUT is used, the residual extra cost of the spheroidal calculation (columns 5-6) compared to the spherical calculation (column 2) is around 15 ns for the small particles and around 20 ns for the largest ones. The effect of the LUT on the accuracy of the calculation (columns 9-10) is below 1%, which is negligible in comparison to the physical uncertainties of the problem.

To compare the efficiency of the method we present here, columns 7-8 give the calculation time to obtain the same result (within an accuracy $\pm 2\%$) with the previously available method (bisection method applied to Eqs. 51-52 of Mallios et al. (2020)). As in Mailler et al. (2023c), large-particle correction is applied only when $\widetilde{R} > 0.0232$, because for smaller $\widetilde{R}$ it changes the value of the settling speed by less than 1%. For particles with $d_{eq} > 10\,\mu m$, large-particle correction is applied, and in this case the calculation time using the explicit expression of $\mathcal{S}$ (Eq. 31) is 3 to 6 times shorter than the explicit resolution of the fixed-point equation. As expected from the results of Mailler et al. (2023c), the difference between the result from the application of Eq. 31 and the explicit resolution of the fixed-point equation by bisection is less than 2% throughout the entire range ($d_{eq} < 10^{-3}\,\mathrm{m}$ and $1 < \lambda < 16$).

## 5 Conclusions

We found that Eq. 25 is valid to express the settling speed of solid aerosol particles in the atmosphere, where function $\mathcal{S}$ depends only on the shape and orientation of the particle. The precise expression of $\mathcal{S}$ is related to the $C_D = f(Re)$ relationship through Eq. 18.

We provide two expressions of $\mathcal{S}$ for prolate spheroids. The first one (Eq. 31) is derived from the theoretical drag formulation of Mallios et al. (2020), and a natural extension of the similar function for spheres (Mailler et al., 2023c). The second one, Eq. 37, with coefficients as expressed in Table. 1, is based on the CFD results of Sanjeevi et al. (2022). Agreement between these two expressions (Fig. 7) is excellent for all prolate spheroids in the atmosphere with $d_{eq} < 10^{-4}\,\mathrm{m}$, with differences below or around 2%. Since the approaches of Mallios et al. (2020) and Sanjeevi et al. (2022) are completely different and independent, these results provide a robust validation of both drag formulations in the Stokes and transition regime.

For higher diameters, differences are still moderate for the horizontally oriented particles (below or around 10%), but much stronger for vertically oriented particles (up and beyond 50%). However, this is not relevant for atmospheric modelling since it has been shown by Mallios et al. (2021) that large atmospheric particles fall in horizontal orientation under the action of the aerodynamic torque. Therefore, the drag formulation based on the results of Mallios et al. (2020), more simple than that of Sanjeevi et al. (2022), applies to all the relevant range for atmospheric particles, so that we propose the following method to estimate the settling speed of prolate aerosols:

1. calculate $r^\phi$ from Eq. 21-22;

2. calculate $C_c^\phi$ from Eq. 20;

3. calculate $\widetilde{U}^{\lambda,\phi}$ from Eq. 24 and Eqs. 8-9;

4. calculate $\widetilde{Ar}$ from Eq. 26;

5. finally calculate $\widetilde{v}_\infty$ from Eq. 25.

AerSett v2.0.2 (Mailler et al. (2023a)) provides a Fortran implementation of these equations, ready-to-use for atmospheric modellers. When large-particle correction is requested, the calculation times obtained with this formulation are 3 to 6 times shorter than previously available methods such as bisection, used in Mallios et al. (2020). We found that storing the output of Equations 8-9 and 21-22 in a precaculated lookup-table permits to reduce the computational time by more than half while changing the numerical results by less than 1%. The present method could be easily extended to particles made of porous materials by considering that a particle made of a material of density $\rho_m$ having porosity $\phi$ can be treated as a dense material with effective density $\rho_p = \rho_m (1 - \phi)$.

From a geophysical point of view and regarding the issue of giant dust, as discussed in Mallios et al. (2021), when the electric field is small or non-existent, such large particles tend to fall with a horizontal orientation ($\phi = \pi/2$) under the effet of the aerodynamic torque, in which case our results indicate that their settling speed would be reduced by about 10% for an aspect ratio $\lambda = 2$, and by about 20% for $\lambda = 4$. This effect may be significant, but is not strong enough to explain the long

atmopheric lifetime of giant dust. However, as shown by Yang et al. (2013), there is observational evidence of shape-induced gravitational sorting during the voyage of dust particles over the Atlantic. Our results give a way to calculate efficiently the shape-induced differences in the settling speed of aerosols, and thereby is one step towards reproducing such effects of shape-induced gravitational sorting in general circulation models or in chemistry-transport models.

From methods based on mechanics and statistical physics, Mallios et al. (2021) have determined the PDFs for particle's attack angle as a function of their aspect ratio and of the other characteristics of the particle and of the fluid (assuming particles shaped as prolate spheroids). Based on these PDFs the authors have calculated the average attack angle of particles with different sizes. They showed that particles with sizes less than $\simeq 2\,\mu\text{m}$ are in principle randomly oriented, while particles with sizes larger than $\simeq 20\,\mu\text{m}$ tend to fall with an essentially horizontal orientation. Therefore, in their present formulation, our results only permit the explicit calculation of the settling speed of giant dust particles with $d_{eq} > 20\,\mu\text{m}$, assuming that their orientation is essentially horizontal. In principle, the results presented here are based on non-dimensional relationships and should be valid also for rigid prolate bodies settling in liquids, in the same ranges of Reynolds tested here (from $Re \ll 1$ to $Re \simeq 300$). In geosciences, this could be of interest for the settling of sediments in lakes or oceans, for example.

A future line of work is to find theoretical and/or heuristic ways to extend our findings to the intermediate orientations and to obtain an expression of the instant settling speed for each possible attack angle. Then, this expression could be integrated on all attack angles (weighted by the PDF of the attack angle) to obtain the resulting average settling speed for a given particle depending on particle's shape and fluid's characteristics, and for all possible sizes of atmospheric aerosols. This shall be the main topic of a future work, which is currently under progress.

Other limitations of the present work include the assumption of prolate spheroidal shape for the dust particles. Taking into account the fact that expressions comparable to Eqs. 8-9 exist for the case of oblate spheroids, we see no particular obstacles in generalizing the approach developed in Mallios et al. (2020) and in the present article to the case of oblate spheroids. The case of triaxial spheroids or of other, more irregular shapes, is still out of reach with the methods developed here.

*Code availability.*  The source code of Aersett v2.0.2 is available online with doi: 10.5281/zenodo.10261378. It is distributed under the GNU General Public Licence v3.0.

All the Python scripts and notebooks used to produce the figures in this manuscript are available at doi: 10.5281/zenodo.11066588. They are distributed under the GNU General Public Licence v3.0.

*Author contributions.*  S. Mailler (SMai), S. Mallios and VA have collaborated to produce the theoretical results of the present study by combining their earlier results in a common framework. SMai has produced the figures. AC, SMai, RP and LM have worked on the implementation of the methods in AerSett v2.0.2. All coauthors have contributed to writing and reviewing the manuscript.

*Competing interests.*  None.

*Acknowledgements.* This study has benefited from the GENCI GEN10274 project for computational resources.

This study has been supported by ADEME (Agence de l'Environnement et de la Maitrise de l'Énergie) under grant ESCALAIR.

This article has been reviewed by Dr. Carlos Alvarez Zambrano and by one anonymous Referee. Authors are grateful to both Referees for their in-depth reading of this manuscript and their useful suggestions.

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
