# Peer review of "New straightforward formulae for the settling speed of prolate spheroids in the atmosphere: theoretical background and implementation in AerSett v2.0.2."

_EGUsphere, 2023_

## Referee Comment (RC3)

Mailler et al: Settling speed of prolate spheroids in the atmosphere

The authors revisit two published results (Mallios et al 2020 and Sanjeevi et al 2022) for the drag coefficient on prolate spheroids in vertical and horizontal orientations settling in an incompressible Newtonian fluid. I would be happy to recommend the manuscript it the authors address my concerns.

Major:

First of all, I am confused whether why the authors need to explicitly calculate the velocity if the drag coefficients are already calculated as functions of two governing dimensionless quantities: Reynolds numbers and aspect ratios. Usually in simulations, equations of motion are made dimensionless and these drag coefficients can then be directly used and there is usually no need of calculating velocities. If the authors can include a justification, that would be better for the readers. Usually in fluid dynamics literature, analyses and calculations are performed in dimensionless forms and as a result one can make use of the drag coefficients themselves and one does not need to worry about the dependence of nondimensional particle velocity and its variation with d_eq. The goal of making governing variables dimensionless is to encode information in a compact form which can then be easily used in calculations. I do not think such an effort to explicitly calculate the "steady state" particle velocity as a function of d_eq is needed in the first place if one solves dimensionless equations of motion.

Instead of presenting their results a functions of d_eq, I urge the authors to first plot the drag coefficients as functions of Re for different \lambda using the results of Mallios et al 2020 and Sanjeevi et al 2022 simultaneously in a single plot. This would clearly showcase the ranges of Re these results can respectively be used and the range for which they are consistent. This eliminates the need to worry about exact values of d_eq.

What are the values of \rho, \rho_P, and \mu did the authors use for their calculations? Does \rho and \mu correspond to values of air? I think these results should equally be valid even in the case when prolate particles are settling in liquids given their Re are in the range when the expressions given by Mallios et al 2020 and Sanjeevi et al 2022 are valid. Why do then the authors focus only on atmosphere?

In addition to Re and \lambda, in the case when Re > 1 Stokes number also becomes important. The authors should comment on why they ignored it. They should also include a discussion addressing the time evolution of particle speed in the cases when particles have a finite Re.

It is well known that prolate particles (spheroidal particles in general see Ardekani et al IJMF 87 2016 PP16-34) rotate as they settle under gravity and attain a steady state orientation such that their broadside is horizontal if their R is lower than a critical value. At R above this critical value, they undergo orientation instability such that they do not have a

steady a steady state orientation. In such a case, how useful are the assumptions about \phi = 0 and \pi/2?

It is well known that the earth's atmosphere has density and viscosity stratification (see Magnaudet and Mercier Annual Review of Fluid Mechanics 2020; More and Ardekani Annual Review of Fluid Mechanics 2023). In such a case how useful are these calculations? Shouldn't one need to include effects of stratification and time dependence then?

Aerosols can also mean liquid droplets suspended in air. The authors should clearly state that by aerosol they strictly focus on solid particles in air as in the case of liquid droplets, the surface boundary conditions are different and the calculations presented are not valid.

Minor:
Eq 15 should have \mu^2 in the denominator. This definition of R is equivalent to something called Archimedes number

Line 233: typo should be "expression"

---

## Author Comment (AC1)

**Answer to Reviewer comment RC1 on the manuscript "**New straightforward formulae for the settling speed of prolate spheroids in the atmosphere: theoretical background and implementation in AerSett v2.0.2.**"**
**(doi: 10.5194/egusphere-2023-2637)**

**January 25, 2024**

We are grateful to Reviewer Carlos Alvarez Zambrano for his careful reading of our manuscript and his insightful questions and suggestions. This document is a contribution to the discussion and not a formal answer in view of the final publication, therefore we will address only the reviewer comments that refer to the scientific content of the manuscript. Reformulations will be addressed for the final submission if we are invited to submit this study to GMD.

**Contents**

**1 Transcript of the Reviewer Comment RC1**

**1.1 Summary**

In this paper, the authors deduced two equations for calculating the settling velocity of atmospheric particles with elongated spheroidal shapes, considering both horizontal and vertical orientations. The first formulation relies solely on theoretical reasoning. The second method is based on drag expressions derived from numerical simulations using computational fluid dynamics (CFD). Their findings indicate that these two formulations yield comparable results, with a deviation, based on the mean particle diameter, within 2% and 10% for particles falling horizontally. The authors also implemented their formulations into a Fortran-based model to calculate dust transport.

**1.2 Overall Evaluation**

The manuscript is well-written, and the authors have done a great job deducing the equations and providing explanations for the reasoning behind them. However, certain sections of the paper, including those related to the formulation deduction, could benefit from additional explanations and discussion. With the incorporation of extra clarifications and/or inclusion of details, in my opinion, this manuscript will ultimately make a good

contribution to the atmospheric dust transport community. Below, I include some questions and comments that could enhance the quality of this paper.

1. I recommend that the authors provide a brief description of AerSett v2.0.2 in the Introduction section, as not everyone may be familiar with this module previously published by (almost) the same authors.

2. Line 67: I suggest changing the expression "might be tricky" to a more formal expression, such as "pose challenges."

3. Line 69: It would be advisable to include the definition of the aspect ratio, even though it is defined later in the document.

4. Abstract and Line 85: It is not clear if the authors implemented both formulations as mentioned in Line 85, or if they used the equation obtained from the first approach, as stated in the Abstract.

5. Equation 10: Define x in the D(x).

6. Equation 11: Is $v_\infty$ the settling velocity for prolate spheroid-shaped particles? If so, what is the main difference with $U^{(\lambda,\phi)}$?

7. Section 2.3: Why is the slip correction factor needed? Is the correction being applied to the whole range of particle sizes? To determine the applicability of the slip-correction factor, the Knudsen number (Kn), the ratio of the mean free path to the particle diameter, needs to be observed. Depending on the calculated value of Kn, the correction may be relevant or not. However, the mean free path depends on the pressure, density, and dynamic viscosity of the air. This raises a question for the authors: do the calculations include variations in these air parameters, or was only a constant pressure considered? I recommend that the authors explore in detail the impact and applicability of the slip correction and include in the paper a discussion of for what particle sizes and/or air pressures the correction is important.

8. Equation 29: Define u in $F_{cg}(u)$.

9. Line 195: The authors state that Eq. 31 provides an accuracy better than 2.5%. However, it is not clear what was the reference used to calculate/compare the results of this equation.

10. Conclusions: I suggest that the authors expand the discussion of the limitations of this formulation. They can explore, for example: i) how other orientation values would change their findings. Although the authors stated that particles tend to fall horizontally, it is also known that during the particle lifespan, they change their orientation. ii) Are there any ideas on how to incorporate porosity into each particle for this new formulation?

**2  Answers**

**2.1  Comment 1. Adding a description of AerSett v2.0.2**

We agree that a description of the module is missing in the introduction, since this module is not (hopefully, not yet) well-known to the community. If we are invited to submit an updated version for final publication we will add this missing piece in the introduction.

**2.2  Comment 4. Did we implement both methods ?**

Line 85 in the manuscript says that "In Section 4 we will present the implementation of both these methods in AerSett v2.0", but Abstract says that "we provide an implementation of the first of these methods in AerSett v2.0.2, a module written in Fortran.". The Reviewer is right in spotting an inconsistency here. The statement in the Abstract is correct, only the first of these method is implemented. In the end of section 3 of the manuscript, we explain why we consider that using the first formulation is more simple and accurate enough for atmospheric sciences.

**2.3  Comment 5. Equation 10: Define x in the D(x).**

Eq. 10 in the manuscript is as follows:

$$C_D\left(Re\right) = \frac{A^{\lambda,\phi}}{Re}\mathcal{D}\left(Re\right), \text{ with } \lim_{x\to 0^+}\mathcal{D}\left(x\right) = 1. \tag{10}$$

In this equation, $x$ is the infinitesimal quantity going to zero in $\lim_{x\to 0^+}\mathcal{D}\left(x\right) = 1$, it is just a dummy variable name. However, introducing a dummy variable here is not indispensable, and may just induce confusion, therefore in case of resubmission we will clarify the meaning by rewriting Eq. 10 as:

$$C_D\left(Re\right) = \frac{A^{\lambda,\phi}}{Re}\mathcal{D}\left(Re\right), \text{ with } \lim_{Re\to 0^+}\mathcal{D}\left(Re\right) = 1, \tag{10}$$

**2.4  Comment 6.**

Eq. 11 and the surrounding text are as follows:

$$v_\infty = \frac{4}{3}\frac{\left(\rho_p - \rho\right)gd_{eq}^2}{A^{\lambda,\phi}\mu\mathcal{D}\left(Re\right)} \tag{11}$$

$$= \frac{U^{\lambda,\phi}}{\mathcal{D}\left(Re\right)}, \tag{12}$$

where $U^{\lambda,\phi} = \frac{4}{3}\frac{\left(\rho_p-\rho\right)gd_{eq}^2}{A^{\lambda,\phi}\mu}$ is the settling velocity of a prolate spheroid with aspect ratio $\lambda$ and orientation angle $\phi$, under the Stokes law for prolate spheroids.

In these equations, $v_\infty$ is the settling velocity for a prolate spheroid-shaped particle, and $\frac{U^{\lambda,\phi}}{\mathcal{D}(Re)}$ is the settling speed of the same particle *under the Stokes law*. More explicitly, $v_\infty$ includes the large-particle drag correction, while $U^{\lambda,\phi}$ does not. Therefore, $U^{\lambda,\phi}$ has an exact analytic expression $U^{\lambda,\phi} = \frac{4}{3}\frac{\left(\rho_p-\rho\right)gd_{eq}^2}{A^{\lambda,\phi}\mu}$, already known from past theoretical works as detailed in the introduction, while $v_\infty$ includes $\mathcal{D}\left(Re\right)$, a drag-correction term that accounts for deviations from the creeping-flow regime that occur for larger Reynolds number.

This distinction will be made more explicit in the manuscript if we are invited to submit a final version of this manuscript.

**2.5  Comment 7. on the slip-correction factor**

We agree that this discussion is important, however it has been done for the case of spherical particles in Mailler et al. (2023) (their Section 5). The conclusions of this figure are not changed in any substantial way for prolate spheroidal particles. In short, the main point-by-point answer to your questions on this point are:

- The slip-correction is needed to take into account the fact that for the smallest particles, their size is comparable to the free mean path of air molecules so that air does not behave like a continuous fluid. We can develop this point in the introduction.

- yes, the correction is applied for the whole range of particle sizes. However, for particles with diameter $D > 10\,\mu$m, this correction is almost negligible (see Fig. 4 of Mailler et al. (2023)).

- Regarding the atmospheric conditions used for this manuscript, only Figures 2 and 5 in the manuscript depend on particular atmospheric conditions. These figures have been produced with $P = 101325\,$Pa and $T = 298.15\,$K. This choice is not specified in the manuscript, we will specify it in a revised version.

- Regarding the influence of atmospheric pressure, temperature and viscosity, Fig. 4 of Mailler et al. (2023) shows that the impact of both the slip-correction and the large-particle drag correction on the settling speed for spherical particles, as a function of particle size and of atmospheric pressure (temperature and viscosity being calculated from pressure using the US Standard Atmosphere).

- We feel that Fig. 4 of Mailler et al. (2023), which is a pressure-diameter diagram, gives an indication as of for which diameters and pressures are slip-correction and/or large-particle drag corrections relevant. We agree that this part of the conclusions of Mailler et al. (2023) needs to be reminded to the Reader in a future version of this manuscript, probably in the introduction. The present manuscript complements this already existing discussion by discussing for which particles eccentricity correction may become substantial (for which we answer in the conclusion that differences begin to be substantial for aspect ration greater than 2).

**2.6 Comment 8. define $u$ in $F_{cg}(u)$**

$u$ is just a dummy variable here, it has no meaning outside of Eq. 29. We will try to rewrite / add a precision at this point if we find a good way to make this clearer.

**2.7 Comment 9. Where does the 2.5% accuracy come from ?**

In line 195 and around, the following statement is made, for which the Reviewers asks for precisions.
" Eq. 18 with $C_D$ as expressed in Eq. 27 yields:

$$\mathcal{S} = (F_{cg}(R \cdot \mathcal{S}))^{-1}. \tag{30}$$

An equivalent fixed-point equation has been solved in Mailler et al. (2023) (their Eqs. 13 and 16), yielding the following approximated expression for $\mathcal{S}(R)$:

$$\mathcal{S}(R) = 1 - \left[1 + \left(\frac{R}{4.880}\right)^{-0.4335}\right]^{-1.905}, \tag{31}$$

which holds with an accuracy better than 2.5% for the $Re < 1000$."
    The justification of this statement is at the core of Mailler et al. (2023), so that we will make it clearer in a future version that the reader is refered to that study for the details of this assertion. The assertion of 2.5% is relative to the loss of accuracy when solving Eq. 30 using explicit expression 31 to obtain the solution right away instead of performing an iterative resolution of Eq. 30.
    We agree that we have to clarify what we mean by "[Eq. 31] holds with an accuracy better than 2.5% for the $Re < 1000$". We do not claim that the accuracy is better than 2.5% relative to real-world data or to an exact theoretical solution (which is not known). We mean that the loss of accuracy in using the explicit formula instead of resolving the fixed-point equation is less than 2.5%, which as we discuss in Mailler et al. (2023) is not a considerable accuracy loss since the Clift-Gauvin formula itself has an uncertainty around 7% compared to real-world measurement and to other comparable formulations (Goossens, 2019).
    We could clarify this by changing sentence in line 195 by: "As discussed in Mailler et al. (2023), using this explicit formula instead of numerically resolving Eq. 30 induces a loss of less than 2.5% in accuracy for $Re < 1000$, which is not critical since, the uncertainty of the Clift-Gauvin formula itself (and of other comparable drag-coefficient formulations) is around 7% when compared to field measurement (Goossens, 2019).

**2.8 Comment 10. Expand the conclusions and discuss the limitations**

We agree that the discussion could be enhanced and in particular the limitations of the present approach could be discussed further. Two points in particular are suggested by the Reviewer.

1. **intermediate orientations** We agree that intermediate orientations have to be dealt with. From methods based on mechanics and statistical physics, Mallios et al. (2021) have determined probability distribution functions (PDFs) for particle's attack angle as a function of their aspect ratio and of the other characteristics of the particle and of the fluid (assuming particles shaped as prolate spheroids). Based on these PDFs the authors have calculated the average attack angle of particles with different sizes. They showed that particles with sizes less than $\simeq 2\,\mu\mathrm{m}$ are in principle randomly oriented, while particles with sizes larger than $\simeq 20\,\mu\mathrm{m}$ tend to fall on average horizontally oriented. A future

line of work is to find theoretical and/or heuristic ways to extend our findings to the intermediate orientations and to obtain an expression of the instant settling speed for each possible attack angle. Then, this expression could be integrated on all attack angles (weighted by the PDF of the attack angle) to obtain the resulting average settling speed for a given particle depending on particle's shape and fluid's characteristics. Further work needs to be done towards this direction, but we can add this as a discussion element, and also discuss how addressing only horizontal and vertical orientations for the moments limits the possible use of our results.

2. **porosity** As long as the particle shape is not affected, porosity can be included easily into our equation system. Let us say that the minerals composing the particle have an overall density $\rho_m$, but has porosity $\phi$. Then, its apparent density $\rho_p$ is $\rho_p = \rho_m \left(1 - \phi\right)$, so that we can use the exact same approach we develop in the manuscript, modulating the value of $\rho_p$ to take porosity into account.

Other limitations include the fact that we have provided expressions for prolate spheroids, but other shapes can occur, in particular oblate spheroids, triaxial spheroids or more irregular shapes. These limitations will be discussed more in-depth as well if we are invited to submit a revised version.

On behalf of the all the authors,

Sylvain Mailler

**References**

Goossens, W. R.: Review of the empirical correlations for the drag coefficient of rigid spheres, Powder Tech., 352, 350–359, https://doi.org/10.1016/j.powtec.2019.04.075, 2019.

Mailler, S., Menut, L., Cholakian, A., and Pennel, R.: AerSett v1.0: a simple and straightforward model for the settling speed of big spherical atmospheric aerosols, Geosci. Model Dev., 16, 1119–1127, https://doi.org/10.5194/gmd-16-1119-2023, 2023.

Mallios, S. A., Daskalopoulou, V., and Amiridis, V.: Orientation of non spherical prolate dust particles moving vertically in the Earth's atmosphere, J. Aerosol Sci., 151, 105 657, https://doi.org/10.1016/j.jaerosci.2020.105657, 2021.

---

## Author Comment (AC2)

**Answer to Reviewer comment RC2 on the manuscript "New straightforward formulae for the settling speed of prolate spheroids in the atmosphere: theoretical background and implementation in AerSett v2.0.2."**

(doi: 10.5194/egusphere-2023-2637)

**March 22, 2024**

We are grateful to Anonymous Reviewer 2 for their careful reading of our manuscript and their insightful questions and suggestions. Their comment seem to be written from a fluid mechanicist point of view, which makes it a particularly useful apport to this discussion, since the manuscript was prepared from an atmospheric physics point of view. We are particularly grateful to the Reviewer to bring to our attention that what we had called the "pseudo-Reynolds number" in the present study and in Mailler et al. (2023) is already known as the Archimedes number. This information will permit us to alleviate and clarify the redaction of our manuscript.

**Contents**

**1 Transcript of the Reviewer Comment RC2**

The authors revisit two published results (Mallios et al., 2020; Sanjeevi et al., 2022) for the drag coefficient on prolate spheroids in vertical and horizontal orientations settling in an incompressible Newtonian fluid. I would be happy to recommend the manuscript it the authors address my concerns.

**1.1 Major**

**1.1.1 Use of dimensionless quantities**

First of all, I am confused whether why the authors need to explicitly calculate the velocity if the drag coefficients are already calculated as functions of two governing dimensionless quantities: Reynolds numbers and aspect ratios. Usually in simulations, equations of motion are made dimensionless and these drag coefficients can then be directly used and there is usually no need of calculating velocities. If the authors can include a justification, that would be better for the readers. Usually in fluid dynamics literature, analyses and calculations are performed in dimensionless forms and as a result one can make use of the drag coefficients themselves and one does not need to worry about the dependence of nondimensional particle velocity and its variation with $d_{eq}$. The goal of making governing variables dimensionless is to encode information in a compact form which can then be easily used in calculations. I do not think such an effort to explicitly calculate the "steady state" particle velocity as a function of $d_{eq}$ is needed in the first place if one solves dimensionless equations of motion.

**1.1.2 Plot drag coefficients as functions of Re**

Instead of presenting their results a functions of $d_{eq}$, I urge the authors to first plot the drag coefficients as functions of Re for different $\lambda$ using the results of Mallios et al. (2020) and Sanjeevi et al. (2022) simultaneously in a single plot. This would clearly showcase the ranges of Re these results can respectively be used and the range for which they are consistent. This eliminates the need to worry about exact values of $d_{eq}$.

**1.1.3 Values of $\rho$, $\rho_p$ and $\mu$. Applicability to liquids.**

What are the values of $\rho$, $\rho_p$, and $\mu$ did the authors use for their calculations? Does $\rho$ and $\mu$ correspond to values of air? I think these results should equally be valid even in the case when prolate particles are settling in liquids given their Re are in the range when the expressions given by Mallios et al. (2020) and Sanjeevi et al. (2022)are valid. Why do then the authors focus only on atmosphere?

**1.1.4 Stokes number**

In addition to Re and $\lambda$, in the case when $Re > 1$, [the] Stokes number also becomes important. The authors should comment on why they ignored it. They should also include a discussion addressing the time evolution of particle speed in the cases when particles have a finite Re.

**1.1.5 Orientation of particles**

It is well known that prolate particles (spheroidal particles in general see Ardekani et al. (2016)) rotate as they settle under gravity and attain a steady state orientation such that their broadside is horizontal if their R is lower than a critical value. At R above this critical value, they undergo orientation instability such that they do not have a steady a steady state orientation. In such a case, how useful are the assumptions about $\phi = 0$ and $\pi/2$?

**1.1.6 Stratification of the atmosphere**

It is well known that the earth's atmosphere has density and viscosity stratification (see Magnaudet and Mercier (2020); More and Ardekani (2023)) In such a case how useful are these calculations? Shouldn't one need to include effects of stratification and time dependence then?

**1.1.7 clearly exclude liquid droplets**

Aerosols can also mean liquid droplets suspended in air. The authors should clearly state that by aerosol they strictly focus on solid particles in air as in the case of liquid droplets, the surface boundary conditions are different and the calculations presented are not valid.

**2  Minor**

1. Eq 15 should have $\mu^2$ in the denominator. This definition of R is equivalent to something called Archimedes number

2. Line 233: typo should be "expression"

**3  Answers**

**3.1  Major**

**3.1.1  Use of dimensionless quantities**

**Comment** "First of all, I am confused whether why the authors need to explicitly calculate the velocity if the drag coefficients are already calculated as functions of two governing dimensionless quantities: Reynolds numbers and aspect ratios. Usually in simulations, equations of motion are made dimensionless and these drag coefficients can then be directly used and there is usually no need of calculating velocities. If the authors can include a justification, that would be better for the readers."

**Answer** Classically, fluid mechanics give the expression of the drag coefficient as a function of Re. In dimensional quantities, this is equivalent to giving the force as a function of the speed, which is enough to solve the equation of motion for the particle. For spheres, several such formulations are discussed in Goossens (2019). For spheroids, Sanjeevi et al. (2022) gives the drag, lift and torque coefficients as a function of Re, of particle aspect ratio and particle orientation.

The reason why this approach is not satisfying for atmospheric science is that the Reynolds number is not known beforehand. Of course, it would be possible to numerically solve the equation of motion for the settling particle until its speed stabilizes, thereby obtaining its terminal fall speed. This would be very time-consuming for atmospheric science in which this calculation would have to be repeated in each model cell and for each possible particle diameter and density. Another, more tractable alternative, is to perform an iterative numerical resolution to calculate the speed as a function of the force. This boils down to the numerical resolution of a non-linear equation, which can be done by dichotomy or any other method, which also has a substantial computational cost.

Another alternative is, as we have done for spherical bodies in Mailler et al. (2023), to use dimensionless quantities to perform this numerical resolution once and for all, and find a direct, approximate expression for the Reynolds number as a function of the Archimedes number. We believe that this approach is particularly suitable for atmospheric sciences for the following reasons:

1. The known parameters of the problem are the size and shape of the particle, its density, and the thermodynamic properties of the carrying fluid (air).

2. What is unknown and needed is the settling speed of the particle, which is an important factor in determining its atmospheric advection and lifetime.

3. Due to the small size of the particles and their lack of inertia, their settling speed is reached almost instantly (compared to their atmospheric lifetime or to the time they need to move towards another atmospheric layer with substantially different characteristics)

**Comment:** Usually in fluid dynamics literature, analyses and calculations are performed in dimensionless forms and as a result one can make use of the drag coefficients themselves and one does not need to worry about the dependence of nondimensional particle velocity and its variation with $d_{eq}$. The goal of making governing variables dimensionless is to encode information in a compact form which can then be easily used in calculations.

**Answer** We agree on the use of non-dimensional variables to "encode information in a compact form". This is why in the present study for spheroids, as in Mailler et al. (2023) for sphere, the method we apply is to build a function giving a non-dimensional speed $\mathcal{S}$ (*a priori* unknown in our atmospheric science problem) as a function of the Archimedes number, known *a priori* in our problem of atmospheric physics (Eq. 31 in the submitted manuscript, for the Mallios et al. (2020) formulation).

Eq. 31 therefore "encodes information in a compact form", and results in Eq. 32 when dimensions are restored (or Eq. 33 if slip-correction is needed).

**Comment:** I do not think such an effort to explicitly calculate the "steady state" particle velocity as a function of $d_{eq}$ is needed in the first place if one solves dimensionless equations of motion.

**Answer:** As said above, we believe that the approach we describe above is minimizing the effort for atmospheric modellers, because we give an explicit formula for the needed quantity (settling velocity) as a function of known quantities (size, shape and density of the particle, thermodynamic properties of the fluid), **without** solving the equation of motion which, as said above, would be too tedious and costly for operational use.

**3.1.2 Plot drag coefficients as functions of Re**

**Comment:** Instead of presenting their results a functions of $d_{eq}$, I urge the authors to first plot the drag coefficients as functions of Re for different $\lambda$ using the results of Mallios et al. (2020) and Sanjeevi et al. (2022) simultaneously in a single plot. This would clearly showcase the ranges of Re these results can respectively be used and the range for which they are consistent. This eliminates the need to worry about exact values of $d_{eq}$.

**Answer:** We agree that such plots would be a more classical way to compare the two approaches in terms $C_D = f(Re)$ profiles. We produced such plots (Figs. 2-1) and will include them and discuss them in the revised version.

Fig. 1 shows that the agreement between both formulations is excellent for small particles (Re=1) , and that a good agreement persists even for big particles in the horizontal orientation (up to Re=300), but for the vertical orientation substantial disagreement arises between the two formulations, with Mallios et al. (2020) giving stronger drag coefficients than Sanjeevi et al. (2022) for this orientation. This is consistent with Fig. 6 in the manuscript, which shows strong differences between both approaches for the vertical orientation and the largest particles (strong Reynolds number). As already noted in the manuscript, the substantial discrepancies between Sanjeevi et al. (2022) and Mallios et al. (2020) in the case of large particles oriented vertically are not a problem because, as mentioned already in the manuscript and by the Reviewer, large particles (with strong Reynolds number) tend to fall with a horizontal orientation.

Figure 2 (which represents $\frac{1}{C_D}$ instead of $C_D$ to avoid masking the substantial differences at high Re) yields the same conclusion, with the additional result that for spherical particles ($\lambda = 1$) the formulae of Mallios et al. (2020) and Sanjeevi et al. (2022) give results that are extremely consistent throughout the range that is relevant for atmospheric aerosol.

**3.1.3 Values of $\rho$, $\rho_p$ and $\mu$. Applicability to liquids.**

**Comment:** What are the values of $\rho$, $\rho_p$, and $\mu$ did the authors use for their calculations? Does $\rho$ and $\mu$ correspond to values of air?

**Answer:** Yes, as also noted by the other Reviewer, these important precisions are missing. The values of $\rho$ and $\mu$ are that of dry air in standard atmospheric conditions ($p = 101325\,\text{Pa}$, $T = 298.15\,\text{K}$). $\rho$ and $\mu$ are calculated from these conditions using the standard preconisations from NOAA/NASA/USAF (1976).

This will be clearly explained in the revised version.

**Comment:** I think these results should equally be valid even in the case when prolate particles are settling in liquids given their Re are in the range when the expressions given by Mallios et al. (2020) and Sanjeevi et al. (2022)are valid. Why do then the authors focus only on atmosphere?

**Answer:** The reason we focus on atmosphere only is subjective, due to the fact that all co-authors work in institutes for atmospheric science. We are not necessarily aware of the possible specificities of other fields. In principle, we agree that the same principles and results should be applicable to particles settling in liquids. In particular, a possible field of application of our work in geophysics could be the settling of particles in lakes and oceans, where the physical problem to solve is comparable (estimate the settling velocity of a particle with known shape, size and density in water with known physical properties). For that, one would need to study the typical shape, size and density of oceanic particles to see what is the typical Reynolds number, and if our method applies. Nevertheless, we will mention this possibility in the Conclusion section of the revised manuscript.

[Figure]

Figure 1: $C_D$ as a function of $\lambda$ for 4 selected values of Re

[Figure]

Figure 2: $C_D$ as a function of Re for 4 selected values of $\lambda$

| D (m) | $\tau$ (s) |
|---|---|
| $10^{-5}$ | $8.0 \times 10^{-4}$ |
| $10^{-4}$ | $8.0 \times 10^{-2}$ |
| $4.5 \times 10^{-4}$ | 1.6 |

Table 1: Response time of the speed of particles as a function of their diameter for $\rho_p = 2650\,\mathrm{kg\,m^{-3}}$, in standard atmospheric conditions ($T = 298.15\,\mathrm{K}$, $p = 101325\,\mathrm{Pa}$)

**3.1.4   Stokes number**

**Comment** In addition to Re and $\lambda$, in the case when $Re > 1$, [the] Stokes number also becomes important. The authors should comment on why they ignored it. They should also include a discussion addressing the time evolution of particle speed in the cases when particles have a finite Re.

   **Answer:** The Stokes number characterizes a particle advected by a fluid flow. It can be defined as:

$$Stk = \frac{\tau \cdot U}{L}, \tag{1}$$

where $\tau$ is a characteristic response time for the particle speed, U a characteristic speed for the flow, and $L$ a characteristic length of the flow. In our system though, it is difficult to define clearly the characteristic length or time for atmospheric flow, and therefore to give a clear expression of the Stokes number. Usually, in atmospheric science, it is considered that, except for the action of gravity, the response time of particles is sufficiently short so that they can be considered to follow passively the trajectories of the air parcels that carry them. Some relevant exceptions to this occur:

1. in the presence of obstacles or vegetation (*e.g.* Pleim et al. (2022)), in which case it is important to determine whether the atmospheric aerosol may be intercepted by vegetation in what is called "dry deposition"

2. in the presence of falling rain drops (*e.g.* Cherrier et al. (2017)), in which case it is important to determine whether the atmospheric aerosol may collide with a raindrop ("below-cloud scavenging").

Other than that, the Stokes number is generally not relevant in the atmospheric science, due to the absence of obstacles in the atmospheric flow.

   However, the question of the Reviewer of the Reviewer regarding whether we have to consider the time evolution of particle speed is relevant. For this, in the Stokes case and for a spherical particle, the response time of the particle is given by:

$$\tau = \frac{\rho_p D^2}{18\mu} \tag{2}$$

   we can calculate $\tau$ as follows for a spherical particle with density $\rho_p = 2650\,\mathrm{kg\,m^{-3}}$ evolving in dry air in standard atmospheric conditions:

   Clearly, all particles with diameter $D < 10^{-4}\,\mathrm{m}$ follow the flow with a lag that is very short relative to any relevant timescale for atmospheric motion (except the previously noted cases of interaction with falling raindrops or with vegetation). $D < 10^{-4}\,\mathrm{m}$ is already extremely large for an atmospheric particle, such particles are scarcely observed in the atmosphere. $4.5 \times 10^{-4}\,\mathrm{m}$ in Table 1 as the maximal value of diameter tested has been chosen because this is the size of the largest atmospheric particle collected by van der Does et al. (2018) in their *in situ* observations of giant dust particles. For such a giant particle, the response time of the particle may become non-negligible in some cases of strong turbulent motion. It is also to be noted that the calculation of $\tau$ done for Table 1 is done assuming a Stokes regime, but the biggest particles strongly deviate from the Stokes regime: the drag force and its derivative with speed become more reduced. Even taking that into account, the reaction time for these extremely large particles particles to adjust to the motion of the flow is at worst of a few seconds. This is short comparable to the typical time-scales of atmospheric motion, and also short compared to the time steps of GCMs or chemistry-transport models (typically, from one minute to a few minutes).

| T (K)   | $\mu$ (Pa s)          |
|---------|-----------------------|
| 273.15  | $1.72 \times 10^{-5}$ |
| 283.15  | $1.77 \times 10^{-5}$ |
| 298.15  | $1.84 \times 10^{-5}$ |

Table 2: Dynamic viscosity of air as a function of temperature

**3.1.5 Orientation of particles**

**Comment:** It is well known that prolate particles (spheroidal particles in general see Ardekani et al. (2016)) rotate as they settle under gravity and attain a steady state orientation such that their broadside is horizontal if their R is lower than a critical value. At R above this critical value, they undergo orientation instability such that they do not have a steady a steady state orientation. In such a case, how useful are the assumptions about $\phi = 0$ and $\pi/2$?

**Answer:** We agree on the observations made by the Reviewer in this comment and the fact that they limit the possible use of our results in their present form. However, results with $\phi = \pi/2$ can be directly used because, as said by the Reviewer and also shown quantitatively in Mallios et al. (2021), the large prolate particles tend to fall with this orientation. This aspect is, in part, already discussed in the manuscript, based on Mallios et al. (2021), which derives a probability distribution function for particle orientation depending on the physical parameters of the problem. For smaller particles, orientation becomes more evenly distributed (and finally random for the smallest ones), as discussed in Mallios et al. (2021).

We are currently working to derive expressions for the settling speed valid for any particular orientation, check them against the Sanjeevi et al. (2022) CFD simulations (which also give data for intermediate orientation), and integrate them over the PDFs of particle orientation given by Mallios et al. (2021). Only this ongoing work will hopefully answer completely and satisfactorily this Reviewer comment.

**3.1.6 Stratification of the atmosphere**

**Comment:** It is well known that the earth's atmosphere has density and viscosity stratification (see Magnaudet and Mercier (2020); More and Ardekani (2023)) In such a case how useful are these calculations? Shouldn't one need to include effects of stratification and time dependence then?

**Answer:** It is true and well-known that the atmosphere is stratified. Density and dynamic viscosity are the key atmospheric variables that affect the settling of particles. Density of air essentially follows an exponential decrease with altitude, with a scale height $H \simeq 8 \times 10^3$ m. This is an extremely smooth evolution.

Dynamic viscosity of air is a direct function of its temperature:

$$\mu = \frac{\beta T^{\frac{3}{2}}}{T + S}, \tag{3}$$

where $\beta = 1.458 \times 10^{-6}$ kg s$^{-1}$ m$^{-1}$ K$^{-\frac{1}{2}}$ and $S = 110.4$ K. The evolution of $\mu$ with temperature for selected T values is shown on Table 2, which clearly shows that even for strong variations of temperature (10 K), $\mu$ varies only by a couple of percent, so even if we suppose a very sharp stratification of the atmosphere where the temperature changes by 10 K over, say, 100 m, the resulting variation in dynamic viscosity will not be substantial: even in the worst case of an extremely big particle with $D = 4.5 \times 10^{-4}$ m, the response time of the particle speed will be a couple of seconds to adjust its settling speed to the environment (see Tab. 1). During this couple of seconds, with a settling speed $U \simeq 6.4$ m s$^{-1}$ obtained with our formulae, the particle will travel over, say, 50 meters, a distance over which the variation of air density is negligible, and that of $\mu$ will be, in the worst case, a couple of percent (Tab. 2). Of course, such a giant diameter is a worst-case only. Particles with $D \simeq 10^{-4}$ m typical of giant dust will have a speed $U \simeq 1.5$ m s$^{-1}$, and during its response time of about 0.1 s (see Table 1) it will travel a dozen of centimeters, a distance across which atmospheric temperature and density does not have relevant variations.

As a result, the vertical scales of atmospheric stratification leave more than enough time for settling particles to adjust rapidly their vertical speed to its steady-state value, even for the biggest atmospheric particles.

**3.1.7 Clearly exclude liquid droplets**

**Comment:** Aerosols can also mean liquid droplets suspended in air. The authors should clearly state that by aerosol they strictly focus on solid particles in air as in the case of liquid droplets, the surface boundary conditions are different and the calculations presented are not valid.

   **Answer:** Many aerosols in the atmosphere are, indeed, liquid. While big hydrometeors can typically be deformed by their interaction with air, liquid aerosol particles tend to be spherical due to surface tension. Liquid particles with a prolate shape have no reason to exist in the atmosphere, and in the discussion of our results we essentially discuss solid particles of mineral dusts. But for the sake of clarity, we will mention clearly that our study is meant only for solid particles.

**3.2 Minor**

**3.2.1 Archimedes number**

**Comment:** Eq 15 should have $\mu^2$ in the denominator. This definition of R is equivalent to something called Archimedes number

   **Answer:** Thank you for having identified this typo. Fortunately, in all the later occurences of this number, the denominator was correctly written with $\mu^2$.

   It is correct that the non-dimensional number that we had defined as "the Reynolds number of a sphere having the same volume as the prolate spheroid and obeying the Stokes law" (under the influence of gravity) and named "pseudo-Reynolds number" in Mailler et al. (2023) and in the present manuscript, is none other than the *Archimedes number*. It is always helpful to name things as they should be named and we are extremely grateful to the Reviewer for permitting us to have this non-dimensional number correctly. Consistently, we will use "Archimedes number" throughout the manuscript, and note it Ar instead of R.

**3.2.2 Typo line 233**

**Answer:** the typo will be corrected in the revised version.

On behalf of the all the authors,

Sylvain Mailler

**References**

Ardekani, M. N., Costa, P., Breugem, W. P., and Brandt, L.: Numerical study of the sedimentation of spheroidal particles, Int. J. Multiphase Flow, 87, 16–34, https://doi.org/10.1016/j.ijmultiphaseflow.2016.08.005, 2016.

Cherrier, G., Belut, E., Gerardin, F., Tanière, A., and Rimbert, N.: Aerosol particles scavenging by a droplet: Microphysical modeling in the Greenfield gap, Atmos. Environ., 166, 519–530, https://doi.org/10.1016/j.atmosenv.2017.07.052, 2017.

Goossens, W. R.: Review of the empirical correlations for the drag coefficient of rigid spheres, Powder Tech., 352, 350–359, https://doi.org/10.1016/j.powtec.2019.04.075, 2019.

Magnaudet, J. and Mercier, M. J.: Particles, Drops, and Bubbles Moving Across Sharp Interfaces and Stratified Layers, Annu. Rev. Fluid Mech., 52, 61–91, https://doi.org/10.1146/annurev-fluid-010719-060139, 2020.

Mailler, S., Menut, L., Cholakian, A., and Pennel, R.: AerSett v1.0: a simple and straightforward model for the settling speed of big spherical atmospheric aerosols, Geosci. Model Dev., 16, 1119–1127, https://doi.org/10.5194/gmd-16-1119-2023, 2023.

Mallios, S. A., Drakaki, E., and Amiridis, V.: Effects of dust particle sphericity and orientation on their gravitational settling in the earth's atmosphere, J. Aerosol Sci., 150, 105 634, https://doi.org/10.1016/j.jaerosci.2020.105634, 2020.

Mallios, S. A., Daskalopoulou, V., and Amiridis, V.: Orientation of non spherical prolate dust particles moving vertically in the Earth's atmosphere, J. Aerosol Sci., 151, 105 657, https://doi.org/10.1016/j.jaerosci.2020.105657, 2021.

More, R. V. and Ardekani, A. M.: Motion in Stratified Fluids, Annu. Rev. Fluid Mech., 55, 157–192, https://doi.org/10.1146/annurev-fluid-120720-011132, 2023.

NOAA/NASA/USAF: U.S Standard Atmosphere 1976, Tech. Rep. NASA-TM-X-74335, NOAA-S/T-76-1562, NASA/NOAA, URL https://hdl.handle.net/11245/1.523366, 1976.

Pleim, J. E., Ran, L., Saylor, R. D., Willison, J., and Binkowski, F. S.: A New Aerosol Dry Deposition Model for Air Quality and Climate Modeling, J. Adv. Model. Earth Sy., 14, e2022MS003 050, https://doi.org/10.1029/2022MS003050, 2022.

Sanjeevi, S. K., Dietiker, J. F., and Padding, J. T.: Accurate hydrodynamic force and torque correlations for prolate spheroids from Stokes regime to high Reynolds numbers, Chemical Engineering Journal, 444, 136 325, https://doi.org/10.1016/j.cej.2022.136325, 2022.

van der Does, M., Knippertz, P., Zschenderlein, P., Giles Harrison, R., and Stuut, J.-B. W.: The mysterious long-range transport of giant mineral dust particles, Sci. Adv., https://doi.org/10.1126/sciadv.aau2768, 2018.

---

## Author Response (AR1)

**Answer to the Reviewers and resubmission of the manuscript "**New straightforward formulae for the settling speed of prolate spheroids in the atmosphere: theoretical background and implementation in AerSett v2.0.1.**"**

**May 2, 2024**

Following the discussion on EGUsphere on our manuscript, we wish to submit the manuscript for *Geosci Model Dev.*. please find below the detailed answers to the Reviewers, and the description of the modifications that have been brought to the text. Following the suggestion of the Reviewers, we have added one new figure panel (Fig. 5c) and two new figures (Figs. 2-3). We also bring more precisions : explanations to several aspects in the presentation and/or description of our results.

Please find below the detailed answers to Reviewer 1 (7 pages) and to reviewer 2 (9 pages).

We hope that with these modifications the manuscript can be for for publication in GMD,

Best regards,

On behalf of the authors,

Sylvain MAILLER

Answer to Reviewer comment RC1 on the manuscript "**New straightforward formulae for the settling speed of prolate spheroids in the atmosphere: theoretical background and implementation in AerSett v2.0.2.**"
(doi: 10.5194/egusphere-2023-2637)

May 2, 2024

We are grateful to Reviewer Carlos Alvarez Zambrano for his careful reading of our manuscript and his insightful questions and suggestions. All the Reviewer comments have been addressed (the corresponding modifications in the manuscript are presented in blue in the present document). In particular, following the Reviewer's request to discuss more the relevance of slip-correction for small particles, we have added a new figure panel to the manuscript (Fig. 5c in the revised manuscript). We hope that with these modifications we have answered satisfactorily all the Reviewer comments.

**Contents**

**1  Transcript of the Reviewer Comment RC1**

**1.1  Summary**

In this paper, the authors deduced two equations for calculating the settling velocity of atmospheric particles with elongated spheroidal shapes, considering both horizontal and vertical orientations. The first formulation relies solely on theoretical reasoning. The second method is based on drag expressions derived from numerical simulations using computational fluid dynamics (CFD). Their findings indicate that these two formulations yield comparable results, with a deviation, based on the mean particle diameter, within 2% and 10% for particles falling horizontally. The authors also implemented their formulations into a Fortran-based model to calculate dust transport.

**1.2  Overall Evaluation**

The manuscript is well-written, and the authors have done a great job deducing the equations and providing explanations for the reasoning behind them. However, certain sections of the paper, including those related to the formulation deduction, could benefit from additional explanations and discussion. With the incorporation of extra clarifications and/or inclusion of details, in my opinion, this manuscript will ultimately make a good

contribution to the atmospheric dust transport community. Below, I include some questions and comments that could enhance the quality of this paper.

1. I recommend that the authors provide a brief description of AerSett v2.0.2 in the Introduction section, as not everyone may be familiar with this module previously published by (almost) the same authors.

2. Line 67: I suggest changing the expression "might be tricky" to a more formal expression, such as "pose challenges."

3. Line 69: It would be advisable to include the definition of the aspect ratio, even though it is defined later in the document.

4. Abstract and Line 85: It is not clear if the authors implemented both formulations as mentioned in Line 85, or if they used the equation obtained from the first approach, as stated in the Abstract.

5. Equation 10: Define x in the D(x).

6. Equation 11: Is $v_\infty$ the settling velocity for prolate spheroid-shaped particles? If so, what is the main difference with $U^{(\lambda,\phi)}$?

7. Section 2.3: Why is the slip correction factor needed? Is the correction being applied to the whole range of particle sizes? To determine the applicability of the slip-correction factor, the Knudsen number (Kn), the ratio of the mean free path to the particle diameter, needs to be observed. Depending on the calculated value of Kn, the correction may be relevant or not. However, the mean free path depends on the pressure, density, and dynamic viscosity of the air. This raises a question for the authors: do the calculations include variations in these air parameters, or was only a constant pressure considered? I recommend that the authors explore in detail the impact and applicability of the slip correction and include in the paper a discussion of for what particle sizes and/or air pressures the correction is important.

8. Equation 29: Define u in $F_{cg}(u)$.

9. Line 195: The authors state that Eq. 31 provides an accuracy better than 2.5%. However, it is not clear what was the reference used to calculate/compare the results of this equation.

10. Conclusions: I suggest that the authors expand the discussion of the limitations of this formulation. They can explore, for example: i) how other orientation values would change their findings. Although the authors stated that particles tend to fall horizontally, it is also known that during the particle lifespan, they change their orientation. ii) Are there any ideas on how to incorporate porosity into each particle for this new formulation?

**2 Answers**

**2.1 Comment 1. Adding a description of AerSett v2.0.2**

We agree that a description of the module was missing in the introduction, since this module is not (hopefully, not yet) well-known to the community. We have added the following sentences into the introduction, to present what was already don in Aersett v1.0, and what will be presented in v2.0.2:

... This formulation has been implemented by the same authors in AerSett v1.0 (Mailler et al., 2023a), a Fortran module designed to be included easily in chemistry-transport models, and already included in Chimere v2023r1 (Menut et al., 2023).

... We will also describe AerSett v2.0.2, a Fortran module designed to calculate accurately and effiently the settling speed of prolate particles oriented either horizontally or vertically in the atmosphere.

**2.2 Comment 4. Did we implement both methods ?**

Line 85 in the manuscript says that "In Section 4 we will present the implementation of both these methods in AerSett v2.0", but Abstract says that "we provide an implementation of the first of these methods in AerSett v2.0.2, a module written in Fortran.". The Reviewer is right in spotting an inconsistency here. The statement in the Abstract is correct, only the first of these method is implemented. In the end of section 3 of the manuscript, we explain why we consider that using the first formulation is more simple and accurate enough for atmospheric sciences.

We have corrected the introdution as follows:

In Section 4 we will present the implementation of the method described in Section 2 in AerSett v2.0.2, and we will give our conclusions in Section 5.

**2.3 Comment 5. Equation 10: Define x in the D(x).**

Eq. 10 in the manuscript is as follows:

$$C_D\left(Re\right) = \frac{A^{\lambda,\phi}}{Re}\mathcal{D}\left(Re\right), \text{ with } \lim_{x\to 0^+}\mathcal{D}\left(x\right) = 1. \tag{10}$$

In this equation, $x$ is the infinitesimal quantity going to zero in $\lim_{x\to 0^+}\mathcal{D}\left(x\right) = 1$, it is just a dummy variable name. However, introducing a dummy variable here is not indispensable, and may just induce confusion, therefore we have clarified Equation 10 as follows:

$$C_D\left(Re\right) = \frac{A^{\lambda,\phi}}{Re}\mathcal{D}\left(Re\right), \text{ with } \lim_{Re\to 0^+}\mathcal{D}\left(Re\right) = 1, \tag{10}$$

**2.4 Comment 6.**

Eq. 11 and the surrounding text are as follows:
"

$$v_\infty = \frac{4}{3}\frac{\left(\rho_p - \rho\right)gd_{eq}^2}{A^{\lambda,\phi}\mu\mathcal{D}\left(Re\right)} \tag{11}$$

$$= \frac{U^{\lambda,\phi}}{\mathcal{D}\left(Re\right)}, \tag{12}$$

where $U^{\lambda,\phi} = \frac{4}{3}\frac{\left(\rho_p-\rho\right)gd_{eq}^2}{A^{\lambda,\phi}\mu}$ is the settling velocity of a prolate spheroid with aspect ratio $\lambda$ and orientation angle $\phi$, under the Stokes law for prolate spheroids."

In these equations, $v_\infty$ is the settling velocity for a prolate spheroid-shaped particle, and $\frac{U^{\lambda,\phi}}{\mathcal{D}(Re)}$ is the settling speed of the same particle *under the Stokes law*. More explicitly, $v_\infty$ includes the large-particle drag correction, while $U^{\lambda,\phi}$ does not. Therefore, $U^{\lambda,\phi}$ has an exact analytic expression $U^{\lambda,\phi} = \frac{4}{3}\frac{\left(\rho_p-\rho\right)gd_{eq}^2}{A^{\lambda,\phi}\mu}$, already known from past theoretical works as detailed in the introduction, while $v_\infty$ includes $\mathcal{D}\left(Re\right)$, a drag-correction term that accounts for deviations from the creeping-flow regime that occur for larger Reynolds number.

We have reformulated the sentence after Eq. 11 as follows:

where $U^{\lambda,\phi} = \frac{4}{3}\frac{\left(\rho_p-\rho\right)gd_{eq}^2}{A^{\lambda,\phi}\mu}$ is the settling velocity of a prolate spheroid with aspect ratio $\lambda$ and orientation angle $\phi$ supposing that the Stokes law is verified exactly. On the other hand, $v_\infty$ is the settling speed of the same prolate spheroid taking into account the deviations from the Stokes law, reflected in the $mathcalD\left(Re\right)$ drag function.

**2.5 Comment 7. on the slip-correction factor**

We agree that this discussion is important, however it has been done for the case of spherical particles in Mailler et al. (2023b) (their Section 5). The conclusions of this figure are not changed in any substantial way for prolate spheroidal particles. In short, the main point-by-point answer to your questions on this point are:

- The slip-correction is needed to take into account the fact that for the smallest particles, their size is comparable to the free mean path of air molecules so that air does not behave like a continuous fluid. We can develop this point in the introduction.

- yes, the correction is applied for the whole range of particle sizes. However, for particles with diameter $D > 10\,\mu\text{m}$, this correction is almost negligible (see Fig. 4 of Mailler et al. (2023b)).

- Regarding the atmospheric conditions used for this manuscript, only Figures 2-3-5 in the initial manuscript (and 4-5-7 in the revised manuscript) depend on particular atmospheric conditions. These figures have been produced with $P = 101325\,\text{Pa}$ and $T = 298.15\,\text{K}$. These precisions have been brought in the revised version of the manuscript (Section 2.4)

- Regarding the influence of atmospheric pressure, temperature and viscosity, Fig. 4 of Mailler et al. (2023b) shows that the impact of both the slip-correction and the large-particle drag correction on the settling speed for spherical particles, as a function of particle size and of atmospheric pressure (temperature and viscosity being calculated from pressure using the US Standard Atmosphere).

- We feel that Fig. 4 of Mailler et al. (2023b), which is a pressure-diameter diagram, gives an indication as of for which diameters and pressures are slip-correction and/or large-particle drag corrections relevant. We agree that this part of the conclusions of Mailler et al. (2023b) needs to be reminded to the Reader in a future version of this manuscript, probably in the introduction. The present manuscript complements this already existing discussion by discussing for which particles eccentricity correction may become substantial (for which we answer in the conclusion that differences begin to be substantial for aspect ration greater than 2).

After our initial answer(previous paragraph), the Reviewer reiterates that:
" I propose that the authors thoroughly investigate the impact and relevance of the slip correction, incorporating a detailed discussion in the paper regarding the particle sizes and/or air pressures for which the correction holds significance. "

A new figure (Fig. 5c) giving the magnitude of slip-correction as a function of particle equivalent diameter and eccentricity has been added, and discussed in the manuscript:
" Figure 5 shows the effect of large-particle correction (Fig. 5a), eccentricity correction effects (Fig. 5b) and slip correction (Fig. 5c) on the settling speed, showing that the large-particle correction begins to be significant ($< -5\%$) for particles with $d_{eq} > 30\,\mu\text{m}$. On the contrary, slip correction is significant ($> 5\%$) only for particles with $d_{eq} < 3 - 5\,\mu\text{m}$, depending on particle eccentricity. For lower pressure values ($p \simeq 200\,\text{hPa}$) representative of the higher troposphere or lower stratosphere, slip correction increases due to the longer mean-free path for air particles in thinner air. At these altitudes, slip-correction reaches 5% for particles with $d_{eq} < 8 - 15\,\mu\text{m}$ (not shown), while large-particle corrections also reaches $-5\%$ for particles with $d_{eq} > 30\,\mu\text{m}$ (not shown).

Figure 1 (not included in the revised manuscript for conciseness) is the same as Fig. 5a-c in the manuscript but for conditions representative of the tropopause following NOAA/NASA/USAF (1976). At these pressure and temperature conditions, we find that slip-correction is significant for $d_{eq} < 8 - 15\,\mu\text{m}$, and large-particle correction for $d_{eq} > 30\,\mu\text{m}$. The conclusions from this additional analysis are mentioned in the revised manuscript (see blue text above).

**2.6 Comment 8. define $u$ in $F_{cg}(u)$**

$u$ is just a dummy variable here, it has no meaning outside of Eq. 29. However, letter $u$ may suggest a speed, in particular in the context of this manuscript, which can mislead the reader.

[Figure]

Figure 1: (a) Large-particle correction $\frac{\widetilde{v}_\infty - \widetilde{U}^{\lambda,\phi}}{\widetilde{U}^{\lambda,\phi}}$ in % (contours); (b) eccentricity correction $\frac{\widetilde{v}_\infty(\lambda;d_{eq}) - \widetilde{v}_\infty(\lambda=0;d_{eq})}{\widetilde{v}_\infty(\lambda=0;d_{eq})}$; and (c) slip-correction $\frac{\widetilde{v}_\infty(\lambda;d_{eq}) - v_\infty(\lambda;d_{eq})}{v_\infty(\lambda;d_{eq})}$. The three panels are in % (contours). The figure is produced for standard atmospheric conditions at the tropopause according to NOAA/NASA/USAF (1976): $(p = 22632.1\,\text{Pa},\ T = 216.65\,\text{K}))$

In the revised version, we have substituted $u$ (which may suggest a speed) by $x$. $x$ is the standard textbook notation for a dummy variable in the definition of a function, which will hopefully reduce the probability of misunderstanding.

**2.7 Comment 9. Where does the 2.5% accuracy come from ?**

In line 195 and around, the following statement is made, for which the Reviewer asks for precisions.
" Eq. 18 with $C_D$ as expressed in Eq. 27 yields:

$$\mathcal{S} = (F_{cg}\,(R \cdot \mathcal{S}))^{-1}. \tag{30}$$

An equivalent fixed-point equation has been solved in Mailler et al. (2023b) (their Eqs. 13 and 16), yielding the following approximated expression for $\mathcal{S}(R)$:

$$\mathcal{S}(R) = 1 - \left[1 + \left(\frac{R}{4.880}\right)^{-0.4335}\right]^{-1.905}, \tag{31}$$

which holds with an accuracy better than 2.5% for the $Re < 1000$."

The justification of this statement is at the core of Mailler et al. (2023b). The assertion of the 2.5% accuracy is to be understood as the loss of accuracy when solving Eq. 30 using explicit expression 31 to obtain the solution right away instead of performing an iterative resolution of Eq. 30.

In the revised manuscript, we have clarified te sentence as follows:
" As discussed in Mailler et al. (2023b), using this explicit formula instead of numerically resolving Eq. 30 induces a loss of less than 2.5% in accuracy for $Re < 1000$, which is not critical since, the uncertainty of the Clift-Gauvin formula itself (and of other comparable drag-coefficient formulations) is around 7% when compared to experimental measurements (Goossens, 2019)."

**2.8 Comment 10. Expand the conclusions and discuss the limitations**

We agree that the discussion could be enhanced and in particular the limitations of the present approach could be discussed further. Two points in particular are suggested by the Reviewer.
**intermediate orientations**

We have added the following piece of text to the manuscript:

[revised manuscript text omitted]

Answer to Reviewer comment RC2 on the manuscript "**New straightforward formulae for the settling speed of prolate spheroids in the atmosphere: theoretical background and implementation in AerSett v2.0.2.**"
(doi: 10.5194/egusphere-2023-2637)

May 2, 2024

We are grateful to Anonymous Reviewer 2 for their careful reading of our manuscript and their insightful questions and suggestions. Their comment seem to be written from a fluid mechanicist point of view, which makes it a particularly useful apport to this discussion, since the manuscript was prepared from an atmospheric physics point of view. We are particularly grateful to the Reviewer to bring to our attention that what we had called the "pseudo-Reynolds number" in the present study and in Mailler et al. (2023) is already known as the Archimedes number. This information will permit us to alleviate and clarify the redaction of our manuscript.

We feel that we have answered all of the Reviewer comments and concerns in the discussion and/or by additions in the revised version of the Manuscript. All the modifications brought to the manuscript following comments by the Reviewer have been highlighted in blue in the present document.

**Contents**

**1 Transcript of the Reviewer Comment RC2**

The authors revisit two published results (Mallios et al., 2020; Sanjeevi et al., 2022) for the drag coefficient on prolate spheroids in vertical and horizontal orientations settling in an incompressible Newtonian fluid. I would be happy to recommend the manuscript it the authors address my concerns.

**1.1 Major**

**1.1.1 Use of dimensionless quantities**

First of all, I am confused whether why the authors need to explicitly calculate the velocity if the drag coefficients are already calculated as functions of two governing dimensionless quantities: Reynolds numbers and aspect ratios. Usually in simulations, equations of motion are made dimensionless and these drag coefficients can then be directly used and there is usually no need of calculating velocities. If the authors can include a justification, that would be better for the readers. Usually in fluid dynamics literature, analyses and calculations are performed in dimensionless forms and as a result one can make use of the drag coefficients themselves and one does not need to worry about the dependence of nondimensional particle velocity and its variation with $d_{eq}$. The goal of making governing variables dimensionless is to encode information in a compact form which can then be easily used in calculations. I do not think such an effort to explicitly calculate the "steady state" particle velocity as a function of $d_{eq}$ is needed in the first place if one solves dimensionless equations of motion.

**1.1.2 Plot drag coefficients as functions of Re**

Instead of presenting their results a functions of $d_{eq}$, I urge the authors to first plot the drag coefficients as functions of Re for different $\lambda$ using the results of Mallios et al. (2020) and Sanjeevi et al. (2022) simultaneously in a single plot. This would clearly showcase the ranges of Re these results can respectively be used and the range for which they are consistent. This eliminates the need to worry about exact values of $d_{eq}$.

**1.1.3 Values of $\rho$, $\rho_p$ and $\mu$. Applicability to liquids.**

What are the values of $\rho$, $\rho_p$, and $\mu$ did the authors use for their calculations? Does $\rho$ and $\mu$ correspond to values of air? I think these results should equally be valid even in the case when prolate particles are settling in liquids given their Re are in the range when the expressions given by Mallios et al. (2020) and Sanjeevi et al. (2022)are valid. Why do then the authors focus only on atmosphere?

**1.1.4 Stokes number**

In addition to Re and $\lambda$, in the case when $Re > 1$, [the] Stokes number also becomes important. The authors should comment on why they ignored it. They should also include a discussion addressing the time evolution of particle speed in the cases when particles have a finite Re.

**1.1.5 Orientation of particles**

It is well known that prolate particles (spheroidal particles in general see Ardekani et al. (2016)) rotate as they settle under gravity and attain a steady state orientation such that their broadside is horizontal if their R is lower than a critical value. At R above this critical value, they undergo orientation instability such that they do not have a steady a steady state orientation. In such a case, how useful are the assumptions about $\phi = 0$ and $\pi/2$?

**1.1.6 Stratification of the atmosphere**

It is well known that the earth's atmosphere has density and viscosity stratification (see Magnaudet and Mercier (2020); More and Ardekani (2023)) In such a case how useful are these calculations? Shouldn't one need to include effects of stratification and time dependence then?

**1.1.7 clearly exclude liquid droplets**

Aerosols can also mean liquid droplets suspended in air. The authors should clearly state that by aerosol they strictly focus on solid particles in air as in the case of liquid droplets, the surface boundary conditions are different and the calculations presented are not valid.

**2 Minor**

1. Eq 15 should have $\mu^2$ in the denominator. This definition of R is equivalent to something called Archimedes number

2. Line 233: typo should be "expression"

**3 Answers**

**3.1 Major**

**3.1.1 Use of dimensionless quantities**

**Comment** "First of all, I am confused whether why the authors need to explicitly calculate the velocity if the drag coefficients are already calculated as functions of two governing dimensionless quantities: Reynolds numbers and aspect ratios. Usually in simulations, equations of motion are made dimensionless and these drag coefficients can then be directly used and there is usually no need of calculating velocities. If the authors can include a justification, that would be better for the readers."

**Answer** Classically, fluid mechanics give the expression of the drag coefficient as a function of Re. In dimensional quantities, this is equivalent to giving the force as a function of the speed, which is enough to solve the equation of motion for the particle. For spheres, several such formulations are discussed in Goossens (2019). For spheroids, Sanjeevi et al. (2022) gives the drag, lift and torque coefficients as a function of Re, of particle aspect ratio and particle orientation.

The reason why this approach is not satisfying for atmospheric science is that the Reynolds number is not known beforehand. Of course, it would be possible to numerically solve the equation of motion for the settling particle until its speed stabilizes, thereby obtaining its terminal fall speed. This would be very time-consuming for atmospheric science in which this calculation would have to be repeated in each model cell and for each possible particle diameter and density. Another, more tractable alternative, is to perform an iterative numerical resolution to calculate the speed as a function of the force. This boils down to the numerical resolution of a non-linear equation, which can be done by dichotomy or any other method, which also has a substantial computational cost.

Another alternative is, as we have done for spherical bodies in Mailler et al. (2023), to use dimensionless quantities to perform this numerical resolution once and for all, and find a direct, approximate expression for the Reynolds number as a function of the Archimedes number. We believe that this approach is particularly suitable for atmospheric sciences for the following reasons:

1. The known parameters of the problem are the size and shape of the particle, its density, and the thermodynamic properties of the carrying fluid (air).

2. What is unknown and needed is the settling speed of the particle, which is an important factor in determining its atmospheric advection and lifetime.

3. Due to the small size of the particles and their lack of inertia, their settling speed is reached almost instantly (compared to their atmospheric lifetime or to the time they need to move towards another atmospheric layer with substantially different characteristics)

We have discussed this problem in the revised version of the introduction by adding the following paragraph:

In the current work, we try to expand this formulation in the case of non spherical solid particles, focusing on prolate spheroids. As in Mailler et al. (2023), the point of this study is to obtain a direct and

computationnally efficient method for the calculation of the settling speed as a function of known parameters (characteristics of the flow and of the particle). This problem is reciprocal of the classical problem in fluid mechanics (calculating the force as a function of the speed). In atmospheric science, the characteristics of the particle, including the gravity force it is submitted to, are known, while the settling speed is not known a priori, making this classical approach impractical for our problem.

**Comment:** Usually in fluid dynamics literature, analyses and calculations are performed in dimensionless forms and as a result one can make use of the drag coefficients themselves and one does not need to worry about the dependence of nondimensional particle velocity and its variation with $d_{eq}$. The goal of making governing variables dimensionless is to encode information in a compact form which can then be easily used in calculations.

**Answer** We agree on the use of non-dimensional variables to "encode information in a compact form". This is why in the present study for spheroids, as in Mailler et al. (2023) for sphere, the method we apply is to build a function giving a non-dimensional speed $\mathcal{S}$ (a priori unknown in our atmospheric science problem) as a function of the Archimedes number, known a priori in our problem of atmospheric physics (Eq. 31 in the submitted manuscript, for the Mallios et al. (2020) formulation).

Eq. 31 therefore "encodes information in a compact form" by giving a mathematical relationship between two non-dimensional quantities - the Archimedes number related to the force, and the $\mathcal{S}$ function related to the speed. This formulation results in Eq. 32 when dimensions are restored (or Eq. 33 if slip-correction is needed).

We feel that replacing the "pseudo-Reynolds number" by its correct name, the well-known Archimedes number, as suggested by the Reviewer, may help the reader to realize that our apprach is essentially based on finding relationships between non-dimensional quantities, as requested by the Reviewer.

**Comment:** I do not think such an effort to explicitly calculate the "steady state" particle velocity as a function of $d_{eq}$ is needed in the first place if one solves dimensionless equations of motion.

**Answer:** As said above, we believe that the approach we describe above is minimizing the effort for atmospheric modellers, because we give an explicit formula for the needed quantity (settling velocity) as a function of known quantities (size, shape and density of the particle, thermodynamic properties of the fluid), **without** solving the equation of motion which, as said above, would be too tedious and costly for operational use. As an illustration of this computational efficiency, Table 2 shows that using the approach developed here for the calculation of the settling speed reduces the calculation time by about a factor 4 (for particles with $D > 10\,\mu\text{m}$ for which a large-particle correction is needed). In the revised version, this is clarified by the following additionnal paragraph (already cited above):

In the current work, we try to expand this formulation in the case of non spherical solid particles, focusing on prolate spheroids. As in Mailler et al. (2023), the point of this study is to obtain a direct and computationnally efficient method for the calculation of the settling speed as a function of known parameters (characteristics of the flow and of the particle). This problem is reciprocal of the classical problem in fluid mechanics (calculating the force as a function of the speed). In atmospheric science, the characteristics of the particle, including the gravity force it is submitted to, are known, while the settling speed is not known a priori, making this classical approach impractical for our problem.

**3.1.2   Plot drag coefficients as functions of Re**

**Comment:** Instead of presenting their results a functions of $d_{eq}$, I urge the authors to first plot the drag coefficients as functions of Re for different $\lambda$ using the results of Mallios et al. (2020) and Sanjeevi et al. (2022) simultaneously in a single plot. This would clearly showcase the ranges of Re these results can respectively be used and the range for which they are consistent. This eliminates the need to worry about exact values of $d_{eq}$.

**Answer:** We agree that such plots are a classical way to compare the two approaches in terms $C_D = f(Re)$ profiles.

We have produced such plots and included them in the revised version of the article (Figs. 2-3). These plots are discussed in Section 2.4 in the revised manuscript.

**3.1.3   Values of $\rho$, $\rho_p$ and $\mu$. Applicability to liquids.**

**Comment:** What are the values of $\rho$, $\rho_p$, and $\mu$ did the authors use for their calculations? Does $\rho$ and $\mu$

correspond to values of air?

**Answer:** Yes, as also noted by the other Reviewer, these important precisions were missing in the initial manuscript. The have been added in the revised version:

Both panels of Fig. 4 as well as all the subsequent figures in the study have been produced using standard atmospheric consitions for air ($p = 101325\,\text{hPa}$ and $T = 298.15\,\text{K}$). Dynamic viscosity $\mu$ has been calculated following the US Standard Atmosphere (NOAA/NASA/USAF, 1976):

$$\mu = \frac{\beta T^{\frac{3}{2}}}{T + S},\tag{1}$$

where $\beta = 1.458 \times 10^{-6}\,\text{kg s}^{-1}\,\text{m}^{-1}\,\text{K}^{-\frac{1}{2}}$ and $S = 110.4\,\text{K}$. In these conditions of temperature and pressure and with the molar mass of dry air $M_a = 28.9644 \times 10^{-3}\,\text{kg mol}^{-1}$ (also from the US Standard Atmosphere), the density of air is $\rho = 1.18\,\text{kg m}^{-3}$.

**Comment:** I think these results should equally be valid even in the case when prolate particles are settling in liquids given their Re are in the range when the expressions given by Mallios et al. (2020) and Sanjeevi et al. (2022)are valid. Why do then the authors focus only on atmosphere?

**Answer:** The reason we focus on atmosphere only is subjective, due to the fact that all co-authors work in institutes for atmospheric science. We are not necessarily aware of the possible specificities of other fields. In principle, we agree that the same principles and results should be applicable to particles settling in liquids. In particular, a possible field of application of our work in geophysics could be the settling of particles in lakes and oceans, where the physical problem to solve is comparable (estimate the settling velocity of a particle with known shape, size and density in water with known physical properties). For that, one would need to study the typical shape, size and density of oceanic particles to see what is the typical Reynolds number, and if our method applies. The following sentences have been added to the conclusions:

In principle, the results presented here are based on non-dimensional relationships and should be valid also for rigid prolate bodies settling in liquids, in the same ranges of Reynolds tested here (from $Re \ll 1$ to $Re \simeq 300$). In geosciences, this could be of interest for the settling of sediments in lakes or oceans, for example.

**3.1.4 Stokes number**

**Comment** In addition to Re and $\lambda$, in the case when $Re > 1$, [the] Stokes number also becomes important. The authors should comment on why they ignored it. They should also include a discussion addressing the time evolution of particle speed in the cases when particles have a finite Re.

**Answer:** The Stokes number characterizes a particle advected by a fluid flow. It can be defined as: 0

$$Stk = \frac{\tau \cdot U}{L},\tag{2}$$

where $\tau$ is a characteristic response time for the particle speed, U a characteristic speed for the flow, and $L$ a characteristic length of the flow. In our system though, it is difficult to define clearly the characteristic length or time for atmospheric flow, and therefore to give a clear expression of the Stokes number. Usually, in atmospheric science, it is considered that, except for the action of gravity, the response time of particles is sufficiently short so that they can be considered to follow passively the trajectories of the air parcels that carry them. Some relevant exceptions to this occur:

1. in the presence of obstacles or vegetation (*e.g.* Pleim et al. (2022)), in which case it is important to determine whether the atmospheric aerosol may be intercepted by vegetation in what is called "dry deposition"

2. in the presence of falling rain drops (*e.g.* Cherrier et al. (2017)), in which case it is important to determine whether the atmospheric aerosol may collide with a raindrop ("below-cloud scavenging").

Other than that, the Stokes number is generally not relevant in the atmospheric science, due to the absence of obstacles in the atmospheric flow.

| D (m) | $\tau$ (s) |
|:---:|:---:|
| $10^{-5}$ | $8.0 \times 10^{-4}$ |
| $10^{-4}$ | $8.0 \times 10^{-2}$ |
| $4.5 \times 10^{-4}$ | $1.6$ |

Table 1: Response time of the speed of particles as a function of their diameter for $\rho_p = 2650 \, \mathrm{kg \, m^{-3}}$, in standard atmospheric conditions ($T = 298.15 \, \mathrm{K}$, $p = 101325 \, \mathrm{Pa}$)

However, the question of the Reviewer regarding whether we have to consider the time evolution of particle speed is relevant. For this, in the Stokes case and for a spherical particle, the response time of the particle is given by:

$$\tau = \frac{\rho_p D^2}{18\mu} \tag{3}$$

we can calculate $\tau$ as follows for a spherical particle with density $\rho_p = 2650 \, \mathrm{kg \, m^{-3}}$ evolving in dry air in standard atmospheric conditions:

Clearly, all particles with diameter $D < 10^{-4} \, \mathrm{m}$ follow the flow with a lag that is very short relative to any relevant timescale for atmospheric motion (except the previously noted cases of interaction with falling raindrops or with vegetation). $D < 10^{-4} \, \mathrm{m}$ is already extremely large for an atmospheric particle, such particles are scarcely observed in the atmosphere. $4.5 \times 10^{-4} \, \mathrm{m}$ in Table 1 as the maximal value of diameter tested has been chosen because this is the size of the largest atmospheric particle collected by van der Does et al. (2018) in their *in situ* observations of giant dust particles. For such a giant particle, the response time of the particle may become non-negligible in some cases of strong turbulent motion. It is also to be noted that the calculation of $\tau$ done for Table 1 is done assuming a Stokes regime, but the biggest particles strongly deviate from the Stokes regime: the drag force and its derivative with speed become more reduced. Even taking that into account, the reaction time for these extremely large particles particles to adjust to the motion of the flow is at worst of a few seconds. This is short comparable to the typical time-scales of atmospheric motion, and also short compared to the time steps of GCMs or chemistry-transport models (typically, from one minute to a few minutes).

**3.1.5 Orientation of particles**

**Comment:** It is well known that prolate particles (spheroidal particles in general see Ardekani et al. (2016)) rotate as they settle under gravity and attain a steady state orientation such that their broadside is horizontal if their R is lower than a critical value. At R above this critical value, they undergo orientation instability such that they do not have a steady a steady state orientation. In such a case, how useful are the assumptions about $\phi = 0$ and $\pi/2$?

**Answer:** We agree on the observations made by the Reviewer in this comment and the fact that they limit the possible use of our results in their present form. However, results with $\phi = \pi/2$ can be directly used because, as said by the Reviewer and also shown in Mallios et al. (2021), the large prolate particles tend fall with this orientation. This aspect is, in part, already discussed in the manuscript, based on Mallios et al. (2021), which derives a probability distribution function for particle orientation depending on the physical parameters of the problem. For smaller particles, orientation becomes more evenly distributed (and finally random for the smallest ones), as discussed in Mallios et al. (2021).

We are currently working to derive expressions for the settling speed valid for any particular orientation, check them against the Sanjeevi et al. (2022) CFD simulations (which also give data for intermediate orientation), and integrate them over the PDFs of particle orientation given by Mallios et al. (2021). Only this ongoing work will hopefully answer completely and satisfactorily this Reviewer comment.

This limitation is now discussed in the Conclusion of the manuscript: A future line of work is to find theoretical and/or heuristic ways to extend our findings to the intermediate orientations and to obtain an expression of the instant settling speed for each possible attack angle. Then, this expression could be integrated on all attack angles (weighted by the PDF of the attack angle) to obtain the resulting average settling speed for a given particle depending on particle's shape and fluid's characteristics, and for all possible

| T (K) | $\mu$ (Pa s) |
|---|---|
| 273.15 | $1.72 \times 10^{-5}$ |
| 283.15 | $1.77 \times 10^{-5}$ |
| 298.15 | $1.84 \times 10^{-5}$ |

Table 2: Dynamic viscosity of air as a function of temperature

sizes of atmospheric aerosols. This shall be the main topic of a future work, which is currently under progress.

**3.1.6 Stratification of the atmosphere**

**Comment:** It is well known that the earth's atmosphere has density and viscosity stratification (see Magnaudet and Mercier (2020); More and Ardekani (2023)) In such a case how useful are these calculations? Shouldn't one need to include effects of stratification and time dependence then?

**Answer:** It is true and well-known that the atmosphere is stratified. Density and dynamic viscosity are the key atmospheric variables that affect the settling of particles. Density of air essentially follows an exponential decrease with altitude, with a scale height $H \simeq 8 \times 10^3$ m. This is an extremely smooth evolution.

Dynamic viscosity of air is a direct function of its temperature:

$$\mu = \frac{\beta T^{\frac{3}{2}}}{T + S},\tag{4}$$

where $\beta = 1.458 \times 10^{-6}$ kg s$^{-1}$ m$^{-1}$ K$^{-\frac{1}{2}}$ and $S = 110.4$ K. The evolution of $\mu$ with temperature for selected T values is shown on Table 2, which clearly shows that even for strong variations of temperature (10 K), $\mu$ varies only by a couple of percent, so even if we suppose a very sharp stratification of the atmosphere where the temperature changes by 10 K over, say, 100 m, the resulting variation in dynamic viscosity will not be substantial: even in the worst case of an extremely big particle with $D = 4.5 \times 10^{-4}$ m, the response time of the particle speed will be a couple of seconds to adjust its settling speed to the environment (see Tab. 1). During this couple of seconds, with a settling speed $U \simeq 6.4$ m s$^{-1}$ obtained with our formulae, the particle will travel over, say, 50 meters, a distance over which the variation of air density is negligible, and that of $\mu$ will be, in the worst case, a couple of percent (Tab. 2). Of course, such a giant diameter is a worst-case only. Particles with $D \simeq 10^{-4}$ m typical of giant dust will have a speed $U \simeq 1.5$ m s$^{-1}$, and during its response time of about 0.1 s (see Table 1) it will travel a dozen of centimeters, a distance across which atmospheric temperature and density does not have relevant variations.

As a result, the vertical scales of atmospheric stratification leave more than enough time for settling particles to adjust rapidly their vertical speed to its steady-state value, even for the biggest atmospheric particles.

**3.1.7 Clearly exclude liquid droplets**

**Comment:** Aerosols can also mean liquid droplets suspended in air. The authors should clearly state that by aerosol they strictly focus on solid particles in air as in the case of liquid droplets, the surface boundary conditions are different and the calculations presented are not valid.

**Answer:** Many aerosols in the atmosphere are, indeed, liquid. While big hydrometeors can typically be deformed by their interaction with air, liquid aerosol particles tend to be spherical due to surface tension. Liquid particles with a prolate shape have no reason to exist in the atmosphere, and in the discussion of our results we essentially discuss solid particles of mineral dusts.

The restriction of this approach to solid aerosols has been mentioned in several places in the revised manuscript (Abstract, Introduction, Conclusion)

**3.2 Minor**

**3.2.1 Archimedes number**

**Comment:** Eq 15 should have $\mu^2$ in the denominator. This definition of R is equivalent to something called

Archimedes number

**Answer:** Thank you for having identified this typo. Fortunately, in all the later occurences of this number, the denominator was correctly written with $\mu^2$.

It is correct that the non-dimensional number that we had defined as "the Reynolds number of a sphere having the same volume as the prolate spheroid and obeying the Stokes law" (under the influence of gravity) and named "pseudo-Reynolds number" in Mailler et al. (2023) and in the initial manuscript, is none other than the *Archimedes number*. It is always helpful to name things as they should be named and we are extremely grateful to the Reviewer for permitting us to have this non-dimensional number correctly.

We use use "Archimedes number" throughout the revised manuscript, and note it Ar instead of R.

**3.2.2 Typo line 233**

**Answer:** Expresison has been corrected to expression

On behalf of the all the authors,

Sylvain Mailler

---

## Author Response (AR2)

**Answer to the Editor after acceptation of the manuscript "**New straightforward formulae for the settling speed of prolate spheroids in the atmosphere: theoretical background and implementation in AerSett v2.0.1.**"**

**May 28, 2024**

Following the acceptation by the Editor of our manuscript and his request for rewording many elements in the text including the title, here are our answers (in blue the Editor comments, in black our answers):

Before we proceed with publication, let me ask for a set of language and technical adjustments:

For Stokes 1851 reference, instead of a .pdf file at a personal homepage, please cite a permanent URL, e.g., https://wellcomecollection.org/works/hcy5wuu4

Done accordingly

line 2: "extremely simple"

We removed "extremely simple"

line 4: "CFD numeric simulations"

We removed CFD

line 38: "managed to increase"

Here, "managed to increase" is factual since one long-standing problem in simulating mineral dust in chemistry-transport model is that models fail to represent the largest particles. The cited paper represented an effort to represent bigger particles, hence the word "manage" here.

line 41: "in general quite computationally expensive"

Replaced by "a computationnally expensive bisection method"

line 44: "managed to improve"

replaced by "improved"

line 58: "a state-of-the-art robust and accurate"

We have removed "state-of-the-art" since this is a recent (2023) publication so the reader can assume it is "state-of-the-art". We removed "robust" because we didn't define robust against what. "Accurate" has a precise meaning which we now develop just before this sentence (and is the main result in the cited study of Mailler 2023). Therefore, we have replaced "state-of-the-art robust and accurate" by "fast and accurate", since for both things we explain just above what we exactly mean with these two adjectives.

line 50: "included easily"

We removed easily

line 51: "we try to expand"

We replaced by "the goal of the present work is to expand..."

line 58: "exact analytical solutions ... give the exact values"

The second occurence of "exact" has been removed.

line 70: "After all," - suggest removing

Has been removed

line 73: "might be tricky, because it can alter the physical result"

Has been replaced by "is important"

line 250: "very differently"

Removed "very"

line 345: typo in "independant"

Fixed

line 349: "more simple"

We kept "more simple" but we specify specified in comparison to what we claim this (comparing to the method exposed in Sanjeevi 2022)

line 394: "SMal" introduced but never used

Removed.

caption of Fig. 5 has a bogus doubled "))"

Fixed.

Presented results are labelled, starting from the title, as "straightforward", "direct", "robust", "simple", "extremely simple", "designed to be included easily", "computationally efficient". I feel that the multitude and vagueness of such wording will not help in convincing readers. Please consider reducing the surplus epithets and matching the wording with supporting statements (if robust, explain to what; if efficient, explain in what way and how faster/smaller; if simple, explain compared to what...).

"straightforward" and "direct" were used to convey the fact that the expressions we give need no iterations or numerical root-finding, avoiding costly calculations. Therefore, we replaced "straigtforward" and "direct" by "explicit" in the entire text, because "explicit" seems to better convey this meaning.

"robust" has been removed except one occurence for which we felt that the meaning is clear.

In the abstract and conclusion; we say that one method is "more simple" than another. This is factual in the sense that one of the two relies on 24 fitted numerical coefficients fed into two different formulae while the other one counts only on one formula with five numerical coefficients for the same office. Therefore, we leave these two instance of "more simple".

The other expressions ("extremely simple", "designed to be included easily", "computationally efficient") have been removed from the revised version.

We hope that with these modifications the manuscript can be fit for for publication in GMD,

Best regards,

On behalf of the authors,

Sylvain MAILLER

[revised manuscript text omitted]